# Zero-shot Concept Bottleneck Models via Sparse Regression of Retrieved Concepts

## Abstract

Concept bottleneck models (CBMs) are inherently interpretable neural network models, which explain their final label prediction by high-level semantic *concepts* predicted in the intermediate layers. Previous works of CBMs have succeeded in achieving high-accuracy concept/label predictions without manually collected concept labels by incorporating large language models (LLMs) and vision-language models (VLMs). However, they still require training on the target dataset to learn input-to-concept and concept-to-label mappings, incurring target dataset collections and training resource requirements. In this paper, we present *zero-shot concept bottleneck models* (Z-CBMs), which are interpretable models predicting labels and concepts in a fully zero-shot manner without training neural networks. Z-CBMs utilize a large-scale concept bank, which is composed of millions of noun phrases extracted from caption datasets, to describe arbitrary input in various domains. To infer the input-to-concept mapping, we introduce *concept retrieval*, which dynamically searches input-related concepts from the concept bank on the multi-modal feature space of pre-trained VLMs. This enables Z-CBMs to handle the millions of concepts and extract appropriate concepts for each input image. In the concept-to-label inference stage, we apply *concept regression* to select important concepts from the retrieved concept candidates containing noisy concepts related to each other. To this end, concept regression estimates the importance weight of concepts with sparse linear regression approximating the input image feature vectors by the weighted sum of concept feature vectors. Through extensive experiments, we confirm that our Z-CBMs achieve both high target task performance and interpretability without any additional training.

## 1 Introduction

One of the primary interests of the deep learning research community is developing a human-interpretable model without performance degradation from black-box deep neural networks. Concept bottleneck model (CBM, Koh et al. (2020)) is an inherently interpretable neural network model, which aims to explain their final prediction via the *concept* predictions in the intermediate layers. CBMs are trained on a target task in an end-to-end manner to learn the input-to-concept and concept-to-label mappings. A concept is composed of high-level semantic vocabulary for describing objects of interest in input data. For instance, CBMs can predict the final label "apple" from the linear combination of the concepts "red sphere," "green leaf," and "glossy surface." In the original CBMs (Koh et al., 2020), a concept set for explaining the prediction is defined by manual annotations for each sample, incurring massive labeling costs greater than ones of the class labels. Another challenge of CBMs is the degradation of target task performance from black-box models due to the long-tailed distribution of the concepts, which is more difficult to learn than the label distribution (Zarlenga et al., 2022). To reduce the costs and maintain the target task performance, Oikarinen et al. (2023) and Yuksekgonul et al. (2023) automatically generate a concept set related to class labels by large language models (LLMs, e.g., GPT-3 (Brown et al., 2020a)) and use the multi-modal embedding space of vision-language models (VLMs, e.g., CLIP (Radford et al., 2021)) to learn the input-to-concept mapping through similarities in the multi-modal feature space. Thanks to the powerful representations of VLMs for mapping input-to-concept, this also alleviates the performance degradation problem of CBMs.

Although modern vision-language-based CBMs are free from manual pre-defined concepts and significant performance degradation, we argue that the practicality is still restricted by the requirements of

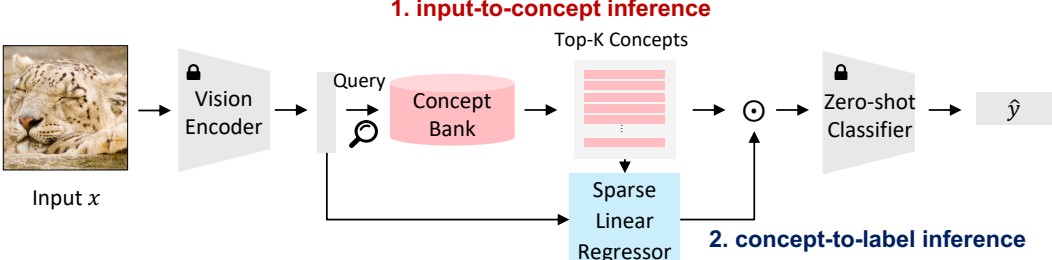

Figure 1: Zero-shot concept bottleneck models (Z-CBMs). Z-CBMs predict concepts for input by retrieving them from a large-scale concept bank. Then, Z-CBMs predict labels based on the weighted sum of the retrieved concept vectors with importance weights yielded by sparse linear regression.

training input-to-concept and concept-to-label mappings on target datasets. In other words, CBMs are not available without manually collecting target datasets and additional training of model parameters on them. To overcome this limitation, this paper tackles a new problem setting of CBMs in a zero-shot manner for target tasks, where we do not assume any target datasets and additional training. In this setting, we can access pre-trained VLMs, but we cannot know the concepts composing target data in advance. This setting forces models to perform two-stage zero-shot inference of input-to-concept and concept-to-label for unseen input samples. The zero-shot input-to-concept inference can not be solved by a naïve application of VLMs as the ordinary zero-shot classification of input-to-label because the concept vocabulary space is much larger than the label space, and the predicted concepts should be a set, not a single label. Furthermore, the zero-shot concept-to-label inference is difficult because the concept-to-label mapping is not obvious without target data and training, which are unavailable in this setting. Therefore, we aim to answer the following research question: *how can we realize the zero-shot inference of CBMs without target datasets and training?*

We present a novel CBM class called *zero-shot concept bottleneck models* (Z-CBMs). Z-CBMs are zero-shot interpretable models that employ off-the-shelf pre-trained VLMs with frozen weights as the backbone. Our key idea is to utilize a large-scale concept set called a concept bank, which is composed of an abundant vocabulary for describing arbitrary input. In contrast to the previous works that deal with only a few thousand concepts at most, our concept bank leverages millions of concepts extracted from large-scale text caption datasets such as YFCC (Thomee et al., 2016) in order to sufficiently cover broad domains for the zero-shot inference. In the input-to-concept inference stage, Z-CBMs dynamically find concept candidates in a concept bank by retrieving them from an input sample in the multi-modal feature space of VLMs (**concept retrieval**). Concept retrieval leverages efficient and scalable similarity search algorithms, e.g., Faiss (Douze et al., 2024; Johnson et al., 2019), allowing Z-CBMs to directly describe concepts with abundant vocabulary without target task training. Then, in the concept-to-label inference stage, Z-CBMs reproduce the zero-shot classification of input-to-label with the backbone VLM by selecting essential concepts from the retrieved concepts. That is, Z-CBMs reconstruct the input visual feature vector by a weighted sum of the concept candidate vectors and then predict the label in the same fashion as the input-to-label zero-shot classification. To reconstruct the vector, we compute the importance weights of the concept candidates by leveraging sparse linear regression such as lasso (**concept regression**). This enables Z-CBMs to naturally select essential concepts from the retrieved concept candidates based on their importance and achieve competitive performance with black-box VLMs.

Our extensive experiments on 12 datasets show that Z-CBMs can achieve competitive performance to backbone VLMs and conventional CBMs. This indicates that the zero-shot inference of Z-CBMs is practical enough for many domains. We also demonstrate that Z-CBMs provide important concepts with their abundant concept vocabulary, which is beyond existing training-based CBMs in terms of the similarity to input images. Furthermore, we show that human experts can intervene in Z-CBMs to improve and analyze the performance through concept deletion/insertion experiments.

## 2 ZERO-SHOT CONCEPT BOTTLLNECK MODELS (Z-CBMS)

We propose Z-CBMs, which first predict interpretable concept candidates from a concept bank composed of abundant vocabulary and then predict the class labels from the weighted sum of predicted concepts (Fig. 1). Unlike conventional CBMs, Z-CBMs can perform a zero-shot inference,

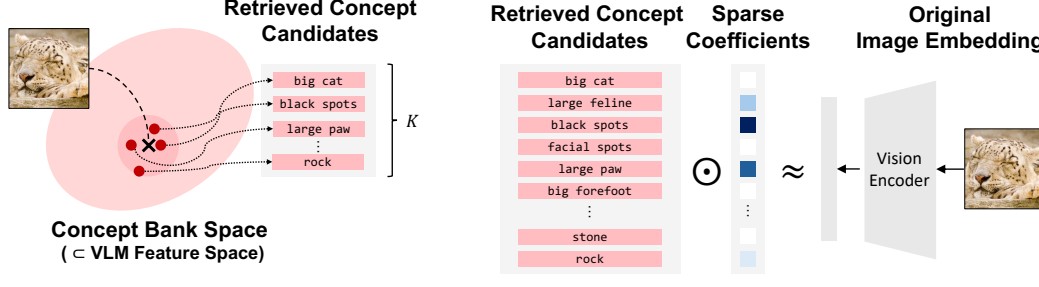

(a) Concept Retrieval  (b) Concept Regression

Figure 2: Concept retrieval and concept regression. (a) Concept retrieval searches concept candidates close to an input image in the VLM feature space and returns the top-$K$ concepts, enabling Z-CBMs to use a large-scale concept bank for general input images. (b) Concept regression selects the important concepts through sparse linear regression, which approximates the input feature vectors by the weighted sum of concept candidate vectors with sparse coefficients. This sparse linear regression is helpful in selecting unique concepts.

i.e., target datasets and additional training are not required. To realize the zero-shot inference, Z-CBMs adopt *concept retrieval* and *concept regression*. Concept retrieval finds a set of the most input-related concept candidates in a concept bank by querying an input image feature with a semantic similarity search (Fig. 2a). Concept regression estimates the importance weights of the concept candidates by sparse linear regression to reconstruct the input feature vector with the weighted sum of concept candidate vectors (Fig. 2b). Finally, Z-CBMs provide the final label predicted by the reconstructed vector and concept explanations with importance scores.

## 2.1 PROBLEM SETTING

We inherit the problem setting of existing vision-language-based CBMs (Oikarinen et al., 2023) except for not updating any neural network parameters. The goal is to predict the final task label $y \in \mathcal{Y}$ of input $x \in \mathcal{X}$ based on $K$ interpretable textual concepts $\{c_i \in \mathcal{C} \subset \mathcal{T}\}_{i=1}^K$, where $\mathcal{X}, \mathcal{Y}, \mathcal{C}$, and $\mathcal{T}$ are the input, label, concept, and text space, respectively. To this end, we predict the final task label by the bi-level prediction $h \circ g(x)$, where $g : \mathcal{X} \to \mathcal{C}^K$ is a concept predictor and $h : \mathcal{C}^K \to \mathcal{Y}$ is a label predictor. This setting allows to access a vision encoder $f_V : \mathcal{X} \to \mathbb{R}^d$ and a text encoder $f_T : \mathcal{T} \to \mathbb{R}^d$ provided by a VLM like CLIP (Radford et al., 2021), and a concept bank $C = \{c_i\}_{i=1}^{N_c}$. The concept bank $C$ is composed of unique concepts from arbitrary sources, including manually collected concepts and automatically generated concepts by LLMs like GPT-3 (Brown et al., 2020a).

## 2.2 ZERO-SHOT INFERENCE

**Concept Retrieval.** We first find the most semantically closed concept candidates to input images from the large spaces in a concept bank (Fig. 2a). Given an input $x$, we retrieve the set of $K$ concept candidates $C_x \subset C$ by using image and text encoders of pre-trained VLMs $f_V$ and $f_T$ as

$$C_x = \text{Ret}_K(f_V(x), f_T(c)) = \underset{c \in C}{\text{Top-K}} \ \text{Sim}(f_V(x), f_T(c)), \qquad (1)$$

where Top-K is an operator yielding top-$K$ concepts in $C$ from a list sorted in descending order according to a similarity metric Sim. Throughout this paper, we use cosine similarity as Sim by following Conti et al. (2023). Thanks to the scalability of the similarity search algorithm (Johnson et al., 2019; Douze et al., 2024), Eq. (1) can efficiently find the concept candidates in an arbitrary concept bank $C$, which contains millions of concepts to describe inputs in various domains.

**Concept Regression.** Given a concept candidate set $C_x = \{c_1, ..., c_K\}$, we predict the final label $\hat{y}$ by selecting essential concepts from $C_x$. Conventional CBMs infer the mapping between $C_x$ and $\hat{y}$ by training neural regression parameters on target tasks, which incurs the requirements of target dataset collections and additional training costs. Instead, we solve this task with a different approach leveraging the zero-shot performance of VLMs. As shown in the previous studies (Radford et al., 2021; Jia et al., 2021), VLMs can be applied to zero-shot classification by inferring a label $\hat{y}$ by

---

**Algorithm 1** Zero-shot Inference of Z-CBMs

---

**Require:** Input $x$, concept bank $C$, image encoder $f_V$, text encoder $f_T$
**Ensure:** Predicted label $\hat{y}$, concepts $C_x$, importance weight $W_{C_x}$
  1: # Retrieving top-K concepts from input
  2: $C_x \leftarrow \underset{c \in C}{\mathrm{Ret}_K}(f_V(x), f_T(c))$
  3: $F_{C_x} \leftarrow [f_T(c_1), ..., f_T(c_K)]$
  4: # Predicting importance weights by sparse linear regression
  5: $W_{C_x} \leftarrow \arg\min_{W \in \mathbb{R}^K} \|f_V(x) - F_{C_x}W\|_2^2 + \lambda\|W\|_1$
  6: # Predicting label by importance weighted sum concept vectors
  7: $\hat{y} \leftarrow \arg\min_{y \in \mathcal{Y}} \mathrm{Sim}(F_{C_x}W_{C_x}, f_T(t_y))$

---

matching input $x$ and a class name text $t_y \in \mathcal{T}$ in the multi-modal feature spaces as follows.

$$\hat{y} = \underset{y \in \mathcal{Y}}{\arg\max} \ \mathrm{Sim}(f_V(x), f_T(t_y)). \tag{2}$$

If the feature vector $f_V(x)$ can be approximated by $C_x$, we can achieve the zero-shot performance of black-box features by interpretable concept features. Based on this idea, we approximate $f_V(x)$ by the weighted sum of the concept features $F_{C_x} = [f_T(c_1), ..., f_T(c_K)] \in \mathbb{R}^{d \times K}$ with an importance weight $W \in \mathbb{R}^K$ (Fig. 2b). To obtain $W$, we solve the linear regression problem defined by

$$\min_W \|f_V(x) - F_{C_x}W\|_2^2 + \lambda\|W\|_1. \tag{3}$$

Through this objective, we can achieve $W$ not only for approximating image features but also for effectively estimating the contribution of each concept to the label prediction owing to the sparse regularization $\|W\|_1$. Since $C_x$ is retrieved from large-scale concept bank $C$, it often contains noisy concepts that are similar to each other, undermining interpretability due to semantic duplication. In this sense, the sparse regularization enhances interpretability since it can eliminate unimportant concepts for the label prediction (Hastie et al., 2015).

**Final Label Prediction.** Finally, we compute the output label with $F_{C_x}$ and $W$ in the same fashion as the zero-shot classification by Eq. (2), i.e.,

$$\hat{y} = \underset{y \in \mathcal{Y}}{\arg\max} \ \mathrm{Sim}(F_{C_x}W, f_T(t_y)). \tag{4}$$

Algorithm 1 shows the overall protocol of the zero-shot inference of Z-CBM. This zero-shot inference algorithm can be applied not only to pre-trained VLMs but also to their linear probing, i.e., fine-tuning a linear head layer on the fixed feature extractor of VLMs for target tasks. We confirm that this simple application is competitive or superior to other vision-language-based CBMs that require additional training of specialized modules in Sec 4.2.

## 3 IMPLEMENTATION

In this section, we present the detailed implementations of Z-CBMs, including backbone VLMs, concept bank construction, concept retrieval, and concept regression.

**Vision-Language Models.** Z-CBMs allow to leverage arbitrary pre-trained VLMs for $f_V$ and $f_T$. We basically use the official implementation of OpenAI CLIP (Radford et al., 2021) and the publicly available pre-trained weights.[1] Specifically, by default, we use ViT-B/32 as $f_V$ and the base transformer with 63M parameters as $f_T$ by following the original CLIP. In Section 4.6.1, we show that other VLM backbones (e.g., SigLIP (Zhai et al., 2023) and OpenCLIP (Cherti et al., 2023)) are also available for Z-CBMs.

**Concept Bank Construction.** Here, we introduce the construction protocols of the concept bank $C$ of Z-CBMs. Since Z-CBMs can not know concepts of input image features in advance, a concept bank should contain sufficient vocabulary to describe the various domain inputs. To this end, we

---

[1] https://github.com/openai/CLIP

extract concepts from multiple image caption datasets and integrate them into a single concept bank. Specifically, we automatically collect concepts as noun phrases by parsing each sentence in the caption datasets including Flickr-30K (Young et al., 2014), CC-3M (Sharma et al., 2018), CC-12M (Changpinyo et al., 2021), and YFCC-15M (Thomee et al., 2016); we use the parser implemented in `nltk` (Bird, 2006). At this time, the concept set size is $|C| \approx$ 20M.

Then, we filter out nonessential concepts from the large base concept set according to several policies. We basically follow the policies introduced by Oikarinen et al. (2023), which removes (i) too long concepts, (ii) too similar concepts to each other, and (iii) too similar concepts to target class names (optional). However, the second policy is computationally intractable because it requires the $\mathcal{O}(|C|^2)$ computation of the similarity matrix across all concepts. Thus, we approximate this using a similarity search by Eq. (1) that yields the most similar concepts. We retrieve the top 64 concepts from a concept and remove them according to the original policy. Finally, after filtering concepts, we obtain the concept bank containing $|C| \approx$ 5M concepts. We also discuss the effect of varying caption datasets used for collecting concepts in Sec. 4.2 and 4.6.2.

**Similarity Search in Concept Retrieval.** Concept retrieval searches the concept candidates from input feature vectors. To this end, we implement the concept search component by the open source library of Faiss (Johnson et al., 2019; Douze et al., 2024). First, we create a search index based on the text feature vectors of all concepts in a concept bank $C$ using $f_T$. At inference time, we retrieve the concept vectors via similarity search on the concept index by specifying the concept number $K$. We found that the choice of $K$ is important because it determines the trade-off between final accuracy and search speed; larger $K$ contributes to finding more effective concepts in concept regression but increases the time for concept retrieval. We set $K = 2048$ as the default value and empirically show the effect of $K$ in Sec. 4.6.

**Sparse Linear Regression in Concept Regression.** In concept regression, we can use arbitrary sparse linear regression algorithms, including lasso (Tibshirani, 1996), elastic net (Zou & Hastie, 2005), and sparsity-constrained optimization like hard thresholding pursuit (Yuan et al., 2014). The efficient implementations of these algorithms are publicly available on the `sklearn` (Pedregosa et al., 2011) and `skscope` (Wang et al., 2024) libraries. The choice of sparse linear regression algorithm depends on the use cases. For example, lasso is useful when one wants to naturally obtain important concepts from a large number of candidate concepts, elastic net is effective for high target task performance, and sparsity-constrained optimization satisfies rigorous requirements regarding the number of concepts for explanations. We use lasso with $\lambda = 1.0 \times 10^{-5}$ as the default algorithm, but we confirm that arbitrary sparse linear regression algorithms are available for Z-CBMs in Sec 4.6.

## 4 EXPERIMENTS

We evaluate Z-CBMs on multiple visual classification datasets and pre-training VLMs. We conduct quantitative experiments on two scenarios: *zero-shot* and *training head*; the former uses pre-trained VLMs for inference without any training, while the latter learns only the classification heads. We also provide qualitative evaluations of output concepts by comparing Z-CBMs with existing vision-language-based CBMs that require additional training.

### 4.1 SETTINGS

**Datasets.** We used 12 image classification datasets containing various image domains: Aircraft (Air) (Maji et al., 2013), Bird (Welinder et al., 2010), Caltech-101 (Cal) (Fei-Fei et al., 2004) Car (Krause et al., 2013), DTD (Cimpoi et al., 2014), EuroSAT (Euro) (Helber et al., 2019), Flower (Flo) (Nilsback & Zisserman, 2008), Food (Bossard et al., 2014), ImageNet (IN) (Russakovsky et al., 2015), Pet (Parkhi et al., 2012), SUN397 (Xiao et al., 2010), and UCF-101 (Soomro, 2012). We use these datasets since they are often used to evaluate the zero-shot generalization performance of VLMs (Radford et al., 2021; Zhou et al., 2022). For the zero-shot scenario, we used the test sets except for ImageNet, and the official validation set for ImageNet. In the training head scenario, we randomly split a training dataset into 9 : 1 and used the former as the training set and the latter as the validation set. For ImageNet, we set the split ratio 99 : 1.

Table 1: Top-1 accuracy on 12 classification datasets with CLIP ViT-B/32.

| Setting | Method | Air | Bird | Cal | Car | DTD | Euro | Flo | Food | IN | Pet | SUN | UCF | Avg. |
|---|---|---|---|---|---|---|---|---|---|---|---|---|---|---|
| Zero-Shot | Zero-shot CLIP | 18.93 | 51.80 | 24.50 | 60.38 | 43.24 | 35.54 | 63.41 | 78.61 | 61.88 | 85.77 | 61.21 | 59.48 | 53.73 |
| | ConSe | 0.99 | 1.87 | 11.68 | 1.42 | 12.23 | 15.32 | 3.51 | 10.99 | 25.19 | 19.16 | 9.65 | 17.76 | 10.82 |
| | Z-CBM (Flickr30K) | 18.27 | 46.70 | 24.26 | 56.46 | 43.56 | 34.32 | 59.80 | 78.17 | 61.52 | 85.46 | 62.23 | 60.67 | 52.62 |
| | Z-CBM (CC3M) | 18.09 | 48.53 | 24.30 | 55.58 | 43.51 | 35.09 | 61.44 | 78.89 | 62.68 | 85.29 | 62.18 | 60.45 | 52.98 |
| | Z-CBM (CC12M) | 18.66 | 51.03 | 24.42 | **59.22** | 43.72 | **36.73** | 63.31 | 79.26 | 62.42 | **85.98** | 62.11 | 60.75 | 52.98 |
| | Z-CBM (YFCC15M) | 18.81 | **51.87** | 24.54 | 58.72 | 43.40 | 35.96 | 63.38 | 79.22 | 62.42 | 85.94 | 62.07 | 60.96 | 53.97 |
| | Z-CBM (ALL) | **19.00** | 51.75 | **25.42** | 58.87 | **43.86** | 36.12 | **63.78** | **82.44** | **62.70** | 85.95 | **62.89** | **61.49** | **54.28** |
| Training Head | Linear Probe CLIP | **45.06** | **72.72** | **95.70** | 79.75 | 74.84 | **92.99** | **94.02** | **87.06** | **68.54** | **88.72** | 65.20 | **83.14** | **78.98** |
| | Label-free CBM | 42.72 | 67.05 | 94.12 | 71.81 | **74.31** | 91.30 | 91.23 | 81.91 | 58.00 | 83.29 | 62.00 | 80.68 | 74.87 |
| | LaBo | 43.43 | 69.38 | 94.82 | 77.78 | 73.59 | 88.17 | 91.67 | 84.29 | 57.70 | 81.26 | 81.26 | 74.04 |
| | CDM | 44.58 | 69.75 | 95.78 | 77.27 | 74.80 | **92.16** | 92.99 | 81.85 | 62.52 | 86.59 | 56.48 | 81.93 | 76.39 |
| | LP-Z-CBM (ALL) | 44.80 | 71.67 | 95.50 | 78.09 | 73.94 | 91.22 | 93.28 | 86.73 | 67.99 | 88.58 | 65.53 | 82.37 | 78.31 |

**Zero-shot Baselines.** Since there are no existing zero-shot baselines of CBMs, we compare our Z-CBMs with the zero-shot inference of a black-box VLM and ConSe (Norouzi et al., 2014) in target task performance. For more details, please see Appendix A.

**Training Head Baselines.** To compare Z-CBMs with existing vision-language-based CBMs, we evaluated models in a relaxed setting where the models are trained on target datasets. In this setting, we applied Z-CBMs to linear probing of VLMs, i.e., fine-tuning only a linear head layer on the feature extractors of VLMs; we refer to this pattern LP-Z-CBM. As the baselines, we used Lable-free CBM (Oikarinen et al., 2023), LaBo (Yang et al., 2023), and CDM (Panousis et al., 2023). We implemented and performed these methods based on their publicly available code repositories.

**Evaluation Metrics.** We report top-1 test accuracy as the target classification task performance. For evaluating predicted concepts, we measured CLIP-Score (Radford et al., 2021; Hessel et al., 2021), which is the cosine similarity between image and text embeddings on CLIP, i.e., higher is better. CLIP-Score between input images and concepts intuitively indicates how well the predicted concept explains the image. Thus, it performs as an indicator to evaluate the quality of the input-to-concept inference. Concretely, we measured averaged CLIP-Scores between test images and the predicted concept texts, where we extracted the top 10 concepts from sorted concepts in descending order by absolute concept importance scores for each model. Furthermore, we used concept coverage to evaluate the Z-CBM's predicted concepts. Concept coverage $|\{c_i^Z\} \cap \{c_i^R\}|/|\{c_i^R\}|$ is the ratio of overlap between Z-CBM's concepts with non-zero coefficients $\{c_i^Z\} \subset C$ and reference concepts $\{c_i^R\} \subset C$ predicted by vision-language-based CBMs that require training. This metric evaluates the extent to which the Z-CBM yields concepts that are close to those derived in the target training when using the shared concept bank $C$. Specifically, we computed the average concept coverage across test samples by using the GPT-generated concept banks by Oikarinen et al. (2023), and reference concepts of Label-free CBMs; we used concepts with contribution scores greater than 0.05 as $\{c_i^R\}$ by following Oikarinen et al. (2023).

## 4.2 ZERO-SHOT INFERENCE ON MULTIPLE DATASETS

Table 1 summarizes the top-1 accuracy for each dataset and the average scores (**Avg.**). It also shows the results when varying the concept bank of Z-CBMs; the brackets in the Z-CBM rows represent the caption dataset used to construct the concept bank. In the zero-shot setting, we surprisingly observed that our Z-CBMs outperformed the zero-shot CLIP baseline in multiple cases (10 of 12 datasets). This may be due to the fact that Z-CBMs approximate image features with the weighted sum of concept text features, reducing the modality gap between the original image and the label text (see Sec. B.1). The ablation study of concept banks demonstrates that higher accuracy tends to be achieved by larger concept banks. This indicates that image features are more accurately approximated by selecting concepts from a rich vocabulary. We further explore the impacts of concept banks in Sec. 4.6.2.

In the training head setting, Z-CBMs based on linear probing models (**LP-Z-CBMs**) reproduced the accuracy of linear probing well. Further, LP-Z-CBMs stably outperformed existing methods that require additional training for special modules. This suggests that our concept retrieval and concept regression using the original CLIP features are sufficient for input-to-concept and concept-to-label inference in terms of target task performance.

Table 2: CLIP-Score on 12 classification datasets with CLIP ViT-B/32. We compute the averaged CLIP-Scores between images and concepts with top-10 absolute coefficients.

| Method | Air | Bird | Cal | Car | DTD | Euro | Flo | Food | IN | Pet | SUN | UCF | Avg. |
|--------|-----|------|-----|-----|-----|------|-----|------|----|-----|-----|-----|------|
| Label-free CBM | 0.6730 | 0.7695 | 0.6934 | 0.7030 | 0.6475 | 0.7310 | 0.6980 | 0.6875 | 0.7056 | 0.7104 | 0.7180 | 0.6580 | 0.6912 |
| LaBo | 0.6817 | 0.7517 | 0.7001 | 0.7197 | 0.6304 | 0.7196 | 0.7063 | 0.7505 | 0.7228 | 0.7031 | 0.7046 | 0.6863 | 0.6980 |
| CDM | 0.6853 | 0.7453 | 0.6958 | 0.7104 | 0.6776 | **0.7359** | 0.7154 | 0.7076 | 0.7445 | 0.7213 | 0.6801 | 0.6928 | 0.7010 |
| Z-CBM (ALL) | **0.7712** | **0.7822** | **0.7693** | **0.7545** | **0.7648** | 0.7323 | **0.7576** | **0.7590** | **0.7746** | **0.7397** | **0.7843** | **0.7751** | **0.7645** |

Table 3: Concept coverage (%) of Z-CBMs on 12 classification datasets with CLIP ViT-B/32.

| Method | Air | Bird | Cal | Car | DTD | Euro | Flo | Food | IN | Pet | SUN | UCF | Avg. |
|--------|-----|------|-----|-----|-----|------|-----|------|----|-----|-----|-----|------|
| Z-CBM (Cosine Similarity) | 66.83 | 41.42 | 37.13 | 60.95 | 71.85 | 90.37 | 50.39 | 77.50 | 48.80 | 90.07 | 29.76 | 37.04 | 58.51 |
| Z-CBM (Linear Regression) | 96.45 | 81.98 | 51.82 | 58.06 | 91.40 | 90.91 | 90.82 | 90.88 | 71.51 | 95.37 | 40.84 | 62.43 | 76.87 |
| Z-CBM (Lasso) | **98.95** | **86.01** | **69.97** | **96.43** | **94.26** | **91.91** | **93.57** | **96.74** | **86.92** | **97.37** | **42.86** | **68.20** | **85.27** |

## 4.3 QUANTITATIVE EVALUATION OF PREDICTED CONCEPTS

Here, we evaluate the predicted concepts of Z-CBMs from the perspective of their factuality to represent image features. For the quantitative evaluation, we measure CLIP-Score and concept coverage across the 12 datasets used in the previous section.

Table 2 shows the results of CLIP-Score. For all datasets, our Z-CBM predicted concepts that are strongly correlated to input images, and it largely outperformed the CBM baselines that require training. This large difference can be caused by the choice of concept bank. Existing CBMs perform concept-to-label inference with learnable parameters, making it difficult to handle millions of concepts at once. Thus, they often limit their concept vocabularies to a few thousand to ensure learnability. In contrast, our Z-CBMs can treat millions of concepts without training by dynamically retrieving concepts of interest and inferring essential concepts with sparse linear regression. That is, paradoxically, Z-CBMs succeed in providing accurate image explanations through an abundant concept vocabulary by eliminating training.

On the other hand, Table 3 shows the results of concept coverage when using the concepts predicted by Label-free CBMs as the reference concepts. We also list the results of Z-CBMs using cosine similarity on CLIP and linear regression to compute the importance coefficients instead of lasso; since all of their coefficients are non-zero values, we measured the concept coverage scores by using the top 128 concepts. Z-CBMs with lasso achieved the best concept coverage; the average score was 85.27%. This indicates that Z-CBMs can predict most of the important concepts found by trained CBMs, and sparse linear regression is a key factor for finding important concepts without training.

## 4.4 QUALITATIVE EVALUATION OF PREDICTED CONCEPTS

Fig. 3 shows the qualitative evaluation of predicted concepts by Label-free CBMs and Z-CBMs with linear regression and lasso when inputting the ImageNet validation examples. Overall, Z-CBMs tend to predict realistic and dominant concepts that appear in input images. For instance, in the first row, Z-CBM predicts various concepts related to dogs, clothes, and background, whereas Label-free CBM focuses on clothes and ignores dogs and background. This difference may be caused by the fact that the image-to-concept mapping of Z-CBMs is not biased toward the label information because it does not train on the target data. Conversely, like the second row, Z-CBMs tend to concentrate on global regions and miss the concepts in local regions; this can be alleviated by intervening the concept prediction (see Sec. 4.5).

For the comparison of linear regression and lasso, we can see that Z-CBM (Linear Reg.) tends to produce concepts that are related to each other. In fact, quantitatively, we also found that the averaged inner CLIP-Scores among the top-10 concepts of lasso is significantly lower than that of linear regression (0.6855 in lasso vs. 0.7826 in linear regression). These results emphasize the advantage of using sparse regression like lasso in concept regression to reduce redundancies of the concepts and to select mutually exclusive concepts based on the concept bank containing abundant vocabulary.

## 4.5 EVALUATION OF HUMAN INTERVENTION

Human intervention in the output concept is an important feature shared by the CBM family for debugging models and modifying the output concepts to make the final prediction accurate. Here, we evaluate the reliability of Z-CBMs through two types of intervention: (i) concept deletion and

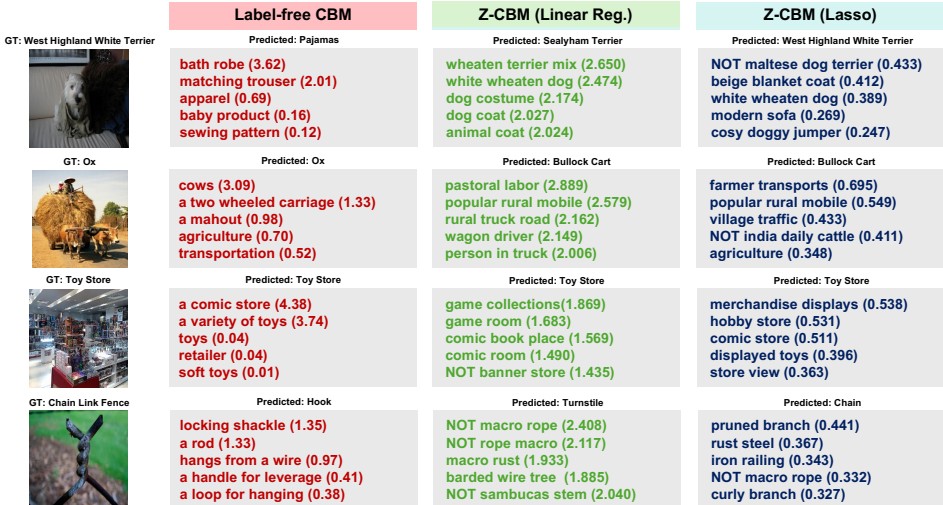

Figure 3: Qualitative evaluation of predicted concepts on the ImageNet validation set. While Label-free CBMs sometimes hallucinate invisible concepts or ignore important concepts, Z-CBMs with lasso consistently provide realistic and dominant concepts in input images with diverse vocabulary. NOT prefix denotes that the concept has negative coefficients.

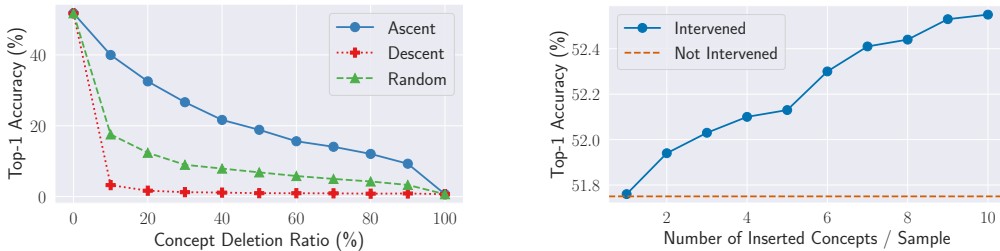

Figure 4: Concept Deletion (Bird)        Figure 5: Concept Insertion (Bird)

(ii) concept insertion. In concept deletion, we confirm the dependence on the predicted concepts by removing the concept with non-zero coefficients in ascending, descending, and random orders. Fig. 4 is the results on Bird by varying the deletion ratio. The accuracy of Z-CBMs significantly dropped with the smaller deletion ratio in the case of descent. This indicates that Z-CBM accurately selects the important concepts through concept regression and predicts the final label based on the concepts. In the case of ascent, the accuracy slowly and steadily decreases, suggesting that the Z-CBMs are not biased toward limited concepts and that all of the selected concepts are essential.

In concept insertion, we add ground truth concepts to the predicted concepts with non-zero coefficients and then re-compute concept regression on the intervened concept set. Specifically, we used linear regression as the algorithm in concept regression and then predicted target labels by the weighted averaged intervened concept vectors by Eq. (4). As the ground truth concepts, we used the fine-grained multi-labels annotated for Bird (Welinder et al., 2010). Fig. 5 demonstrates the top-1 accuracy of the intervened Z-CBMs. The performance improved as the number of inserted concepts per sample increased. This indicates that Z-CBMs can correct the final output by modifying the concept of interest through intervention.

## 4.6 DETAILED ANALYSIS

### 4.6.1 EFFECTS OF BACKBONE VLMS

We show the impacts on Z-CBMs when varying backbone VLMs. Since vision-language models are being intensively studied, it is important to confirm the compatibility of Z-CBMs with successor models with better zero-shot performance. In addition to the CLIP models, we used OpenCLIP (Cherti et al., 2023), SigLIP (Zhai et al., 2023), and DFN (Fang et al., 2024). Table 4 demonstrates the

Table 4: Performance of Z-CBMs varying backbone VLMs on ImageNet.

| Backbone VLM | Top-1 Acc. (Black Box) | Top-1 Acc. (Z-CBM) | CLIP-Score (Z-CBM) |
|---|---|---|---|
| CLIP ViT-B/32 | 61.88 | 62.70 | 0.7746 |
| CLIP ViT-L/14 | 72.87 | 73.19 | 0.7856 |
| OpenCLIP ViT-H/14 | 77.20 | 77.81 | 0.7860 |
| OpenCLIP ViT-G/14 | 79.03 | 78.27 | 0.8049 |
| SigLIP ViT-SO400M/14 | 82.27 | 81.74 | 0.8123 |
| DFN ViT-H/14 | 83.85 | 83.40 | 0.8240 |

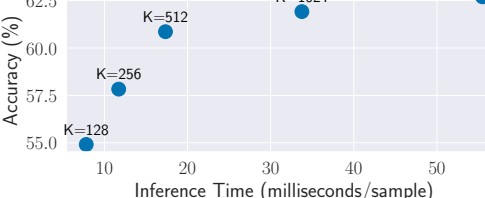

Figure 6: Accuracy vs. inference time by varying retrieved concept number $K$.

Table 5: Performance of Z-CBMs varying concept banks on ImageNet with CLIP ViT-B/32.

| Concept Bank | Vocab. Size | Top-1 Acc. | CLIP-Score |
|---|---|---|---|
| Zero-shot CLIP | N/A | 61.88 | N/A |
| Label-free CBM w/ GPT-3 (ImageNet Class) | 4K | 58.00 | 0.7056 |
| CDM w/ GPT-3 (ImageNet Class) | 4K | 62.52 | 0.7445 |
| GPT-3 (ImageNet Class) | 4K | 59.18 | 0.6276 |
| Noun Phrase (Flickr30K) | 45K | 61.52 | 0.6770 |
| Noun Phrase (CC3M) | 186K | 62.38 | 0.7109 |
| Noun Phrase (CC12M) | 2.58M | 62.42 | 0.7671 |
| Noun Phrase (YFCC15M) | 2.20M | 62.45 | 0.7679 |
| Noun Phrase (ALL) | 5.12M | 62.70 | 0.7746 |

Table 6: Performance of Z-CBMs varying concept regressor on ImageNet with CLIP ViT-B/32.

| Concept Regressor | Top-1 Acc. | Sparsity | CLIP-Score |
|---|---|---|---|
| CLIP Similarity | 14.66 | 0.0000 | 0.8117 |
| Linear Regression | 52.88 | 0.0000 | 0.7076 |
| Lasso | 62.70 | 0.8201 | 0.7746 |
| Elastic Net | 62.84 | 0.7311 | 0.7818 |
| Sparsity-Constrained (HTP) | 62.54 | 0.8750 | 0.7795 |

results, including the original zero-shot classification accuracy and the accuracy with Z-CBMs, and CLIP-Score. The performance of Z-CBMs improved in proportion to the zero-shot performance of the VLMs. In particular, the gradual improvement in CLIP-Score indicates that input-to-concept inference becomes more accurate with more powerful VLMs. These results suggest that Z-CBM is universally applicable across generations of VLMs, and that its practicality will improve as VLMs evolve in future work.

### 4.6.2 EFFECTS OF CONCEPT BANK

As shown in Sec. 4.2 and Table 1, the choice of concept bank is crucial for the performance. Here, we provide a more detailed analysis of the concept banks. Table 5 summarizes the results when varying concept banks. For comparison, we added the concept bank generated by GPT-3 from ImageNet class names, which is used in Label-free CBMs (Oikarinen et al., 2023); we used the concept sets published in the official repository. Although it is competitive with the existing CBM baseline (Label-free CBMs), Z-CBMs with the GPT-3 concepts significantly degraded the top-1 accuracy from Zero-shot CLIP, and the CLIP score was much lower than that of our concept banks composed of noun phrases extracted from caption datasets. This indicates that the concept bank used in the existing method is limited in its ability to represent image concepts. Meanwhile, our concept bank scalably improved in accuracy and CLIP-Score as its size increased, and combining all of them achieved the best results.

### 4.6.3 EFFECTS OF $K$ IN CONCEPT RETRIEVAL

As discussed in Sec. 3, the retrieved concept number $K$ in concept retrieval controls the trade-off between the accuracy and inference time. We assess the effects of $K$ by varying it in $[128, 256, 512, 1024, 2048]$ and measuring the top-1 accuracy and averaged inference time for processing an image. Note that we set 2048 as the maximum value of $K$ because it is the upper bound in the GPU implementation of Faiss (Johnson et al., 2019). Figure 6 illustrates the relationship between the accuracy and total inference time. As expected, the size of $K$ produces a trade-off between accuracy and inference time. Even so, the increase in inference time with increasing $K$ is not explosive and is sufficiently practical since the inferences can be completed in around 55 milliseconds per sample. The detailed breakdowns of total inference time when $K = 2048$ were 0.11 for extracting image features, 5.35 for concept retrieval, and 49.23 for concept regression, indicating that the computation time of concept regression is dominant for the total. In future work, we explore speeding up methods for Z-CBMs to be competitive with the existing CBMs baseline that require training (e.g., Label-free CBMs, which infer a sample in 3.30 milliseconds).

### 4.6.4 EFFECTS OF CONCEPT REGRESSOR

Z-CBMs allow users to choose arbitrary sparse linear regression algorithms according to their demands, as discussed in Sec. 3. Here, we compare the performance of Z-CBMs with multiple

sparse linear regression algorithms: lasso (Tibshirani, 1996), elastic net (Zou & Hastie, 2005), and sparsity-constrained optimization with HTP (Yuan et al., 2014). Further, we evaluate these sparse algorithms by comparing them with non-sparse algorithms to compute the importance of concepts: CLIP Similarity, which uses the cosine similarity computed on CLIP as the importance, and linear regression. Table 6 shows the performance, where sparsity is a ratio of non-zero importance coefficients to the total number of concept candidates. While the sparse linear regression algorithms achieved top-1 accuracy scores at the same level, the non-sparse algorithms failed to accurately predict labels from importance-weighted concepts. Additionally, linear regression has unstable numerical computation due to the rank-deficient of the Gram matrix of $F_{C_x}$ when the feature dimension $d$ is smaller than the concept retrieval size $K$. In contrast, lasso can avoid this by sparse regularization. These results indicate that the concept selection by sparse linear regression is crucial in Z-CBMs. In this sense, we can interpret our concept regression as a re-ranking method of the CLIP similarity. Elastic net was the best in terms of accuracy, but it selected more concepts than the other sparse algorithms. This is because elastic net selects all highly correlated concepts to derive a unique solution by combining $\ell_1$ and $\ell_2$ regularization (Hastie et al., 2015). HTP explicitly limits the number of concepts selected to 256, so while it achieves the highest sparsity, it had the lowest accuracy of the sparse algorithms due to the shortage of concepts for explanation.

## 5 RELATED WORK

CBMs (Koh et al., 2020) are inherently interpretable deep neural network models that predict concept labels and then predict final class labels from the predicted concepts. In contrast to the other explanation styles such as post-hoc attribution heatmaps (Lundberg & Lee, 2017; Selvaraju et al., 2017; Sundararajan et al., 2017), CBMs provide semantic ingredients consisting the final label prediction through the bilevel prediction of input-to-concept and concept-to-label. The original CBMs have the challenge of requiring human annotations of concept labels, which are more difficult to obtain than target task labels. Another challenge is the performance degradation from backbone black-box models (Zarlenga et al., 2022; Moayeri et al., 2023; Xu et al., 2024) due to the difficulty of learning long-tailed concept distributions (Ramaswamy et al., 2023). Post-hoc CBMs (Yuksekgonul et al., 2023), Label-free CBMs (Oikarinen et al., 2023), and LaBo (Yang et al., 2023) addressed these challenges by automatically collecting concepts corresponding to target task labels by querying LLMs (e.g., GPT-3 Brown et al. (2020b)) and leveraging multi-modal feature spaces of pre-trained VLMs (e.g., CLIP Radford et al. (2021)) for learning the input-to-concept mapping. Subsequently, the successor works have basically assumed the use of LLMs or VLMs, further advancing CBMs (Panousis et al., 2023; Rao et al., 2024b; Tan et al., 2024; Srivastava et al., 2024). In particular, Panousis et al. (2023) and Rao et al. (2024a) are related to our work in terms of using space modeling to select concepts for input images. However, all of these existing CBMs still require training specialized neural networks on target datasets, incurring additional target data collection and training resources. Furthermore, these CBMs limit the number of concepts up to a few thousand due to training constraints, restricting the generality. In contrast to the previous CBMs, our Z-CBMs can perform fully zero-shot inference based on a large-scale concept bank with millions of vocabulary for arbitrary input images in various domains as shown in the experiments in Sec. 4.2.

## 6 CONCLUSION

In this paper, we presented zero-shot CBMs (Z-CBMs), which predict input-to-concept and concept-to-label mappings in a fully zero-shot manner. To this end, Z-CBMs first search input-related concept candidates by concept retrieval, which leverages pre-trained VLMs and a large-scale concept bank containing general concepts to describe arbitrary input images in various domains. For the concept-to-label inference, concept regression estimates the importance of concepts by solving the sparse linear regression approximating the input image features by linear combinations of concepts. Our extensive experiments show that Z-CBMs can achieve performance comparable to black-box VLMs and provide interpretable concepts comparable to conventional CBMs that require training. Furthermore, we observed that in some cases, representing image features as linear combinations of concepts reduces the domain gap with label prompts and improves the zero-shot performance. Since Z-CBMs can be built on any off-the-shelf VLMs, we believe that it will be a good baseline for zero-shot interpretable models based on VLMs in future research.

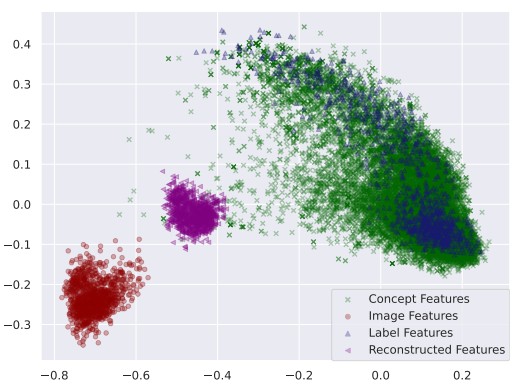 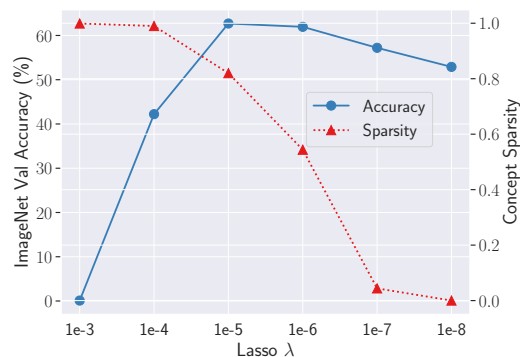

Figure 7: PCA feature visualization of Z-CBMs          Figure 8: Effects of varying $\lambda$ in Eq. 3

## A    DETAILS OF SETTINGS

**Zero-shot Baselines.**    For the black-box baseline, according to the previous work (Radford et al., 2021), we construct a class name prompt $t_y$ by the scheme of "`a photo of [class name]`", and make VLMs predict a target label $\hat{y}$ by Eq. (2). ConSe is a zero-shot cross-modal classification method that infers a target label from a semantic embedding composed of the weighted sum of concepts of the single predicted ImageNet label. We implemented ConSe with pre-trained CLIP and concept bank, which were the same as Z-CBMs. For Z-CBMs, we selected $1.0 \times 10^{-5}$ as $\lambda$ by searching from $\{1.0 \times 10^{-2}, 1.0 \times 10^{-3}, 1.0 \times 10^{-4}, 1.0 \times 10^{-5}, 1.0 \times 10^{-6}, 1.0 \times 10^{-7}, 1.0 \times 10^{-8}\}$ to choose the minimum value achieving over 10% non-zero concept ration when using $K = 2048$ on the subset of ImageNet training set. We used the same $\lambda$ for all experiments.

## B    ADDITIONAL EXPERIMENTS

### B.1    ANALYSIS ON MODALITY GAP

In Section 4.2, Table 1 shows that Z-CBMs improved the zero-shot CLIP baselines. We hypothesize that the reason is reducing the modality gap (Liang et al., 2022) between image and text features by the weighted sum of concept features to approximate $f_V(x)$ by Eq. 3. To confirm this, we conduct a deeper analysis of the effects of Z-CBMs on the modality gap with quantitative and qualitative evaluations. For quantitative evaluation, we measured the L2 distance between image-label features and concept-label features as the modality gap by following (Liang et al., 2022). The L2 distances were $1.74 \times 10^{-3}$ in image-to-label and $0.86 \times 10^{-3}$ in concept-to-label, demonstrating that Z-CBMs largely reduce the modality gap by concept regression. We also show the PCA feature visualizations in Figure 7, indicating that the weighted sums of concepts (reconstructed concepts) bridge the image and text modalities.

### B.2    EFFECTS OF $\lambda$

Here, we discuss the effects when changing $\lambda$ in Eq. (3). We varied $\lambda$ in $\{1.0 \times 10^{-2}, 1.0 \times 10^{-3}, 1.0 \times 10^{-4}, 1.0 \times 10^{-5}, 1.0 \times 10^{-6}, 1.0 \times 10^{-7}, 1.0 \times 10^{-8}\}$. Figure 8 plots the accuracy and the sparsity of predicted concepts on ImageNet. Using different lambda varies the sparsity and accuracy. Therefore, selecting appropriate $\lambda$ is important for achieving both high sparsity and high accuracy.

## C    EXTENDED RELATED WORK

**Cross-modal zero-shot classification.**    In zero-shot or supervised learning settings, several works (Lampert et al., 2013; Norouzi et al., 2014; Mensink et al., 2014; Jain et al., 2015; Elhoseiny et al., 2013) have explored cross-modal classification methodologies by using textual attributes/concepts as a proxy of image features. ConSe (Norouzi et al., 2014) infers a target label from a semantic embedding composed of a weighted sum of concepts of the single predicted ImageNet label with

word2vec embeddings in a fully zero-shot manner. While ConSe is conceptually similar to our Z-CBMs, the zero-shot inference depends on the ImageNet label space, i.e., it cannot accurately predict target labels if there are no target-related labels in ImageNet. In contrast, our Z-CBMs directly decompose an input image feature into concepts via a concept bank, so they are not restricted to any external fixed-label spaces. As a successor work of ConSe, A2C (Demirel et al., 2017) learns input-to-attribute and attribute-to-label mapping by using attributed image datasets for zero-shot inference. While A2C succeeds in outperforming ConSe, the concepts to represent images are restricted to the training datasets, whereas our Z-CBMs are available without additional training and datasets. More recently, Menon & Vondrick (2023) proposed a zero-shot classification method based on the correlation between the input features and the task-specialized texts generated by LLMs for each target class. However, it requires generating the task-specialized texts with LLM and restricting the inference algorithm to the CLIP style zero-shot classification. In contrast, Z-CBMs can be used for arbitrary tasks without external LLMs and arbitrary inference algorithms (e.g., linear probing).

**Ethics Statement.** A potential ethical risk of our proposed method is the possibility that biased vocabulary contained in the concept bank may be output as explanations. Since the concept bank is automatically generated from the caption dataset, it should be properly pre-processed using a filtering tool such as Detoxify (Hanu & Unitary team, 2020) if the data source can be biased.

**Reproducibility Statement.** As described in Sec. 3 and 4 , the implementation of the proposed method uses a publicly available code base. For example, the VLMs backbones are publicly available in the OpenAI CLIP[2] and Open CLIP[3] GitHub repositories. All datasets are also available on the web; see the references in Sec. 4.1 for details. For the computation resources, we used a 24-core Intel Xeon CPU with an NVIDIA A100 GPU with 80GB VRAM. More details of our implementation can be found in the attached code in the supplementary materials and we will make the code available on the public repository if the paper is accepted.

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
