# OpenReview forum: "Zero-shot Concept Bottleneck Models via Sparse Regression of Retrieved Concepts"
_ICLR.cc/2025/Conference — Submitted to ICLR 2025_

### Official Review · Reviewer_mgBg · 2024-10-16

**Soundness:** 3
**Presentation:** 3
**Contribution:** 3
**Rating:** 5
**Confidence:** 5

**Summary:**

This paper proposed a novel approach for achieving zero-shot image classification based on explainable Concept Bottlenecks. Compared with existing Concept Bottleneck Models, the authors proposed approach gets rid of the requirement of labelled training data for learning the mapping network from concepts to categories by fitting the image representation with concept features. The experimental results verified the effectiveness of this approach.

**Strengths:**

This paper provides a novel interpretable zero-shot image classification method.

Compared with existing Concept Bottleneck Models, the proposed method eliminates the requirement of labeled training data.

This paper provides a tool for researchers to understand the semantics of CLIP-extracted visual features.

**Weaknesses:**

The inference cost is significantly increased due to the extremely large concept bank and the test-time learning process.

This paper lacks discussion of other training-free concept bottleneck approaches, e.g., “Visual Classification via Description from Large Language Models”.

**Questions:**

The authors may consider comparing the inference speed between the proposed approach and existing CBMs.

I’m wondering whether the visual and textual features are in the same space as shown in Fig 2 (a) for fitting image features with textual features of candidate concepts, considering that they are from two modalities and in the pre-training stage, the text features and visual features are aligned by cross-entropy loss rather than strictly calibrated by L2 Loss. The authors may consider showing a t-SNE figure to clarify this.

The authors may consider evaluating the interpretability of the candidate concepts. In my opinion, concepts such as “Not maltese dog terrier” cannot provide interpretable information for identifying categories.

---

> ### Author Response · Authors · 2024-11-19
> **Author Response for mgBg**
>
> Thank you for your positive and valuable feedback. We address your concerns bellow. We will revise our paper according to your suggestions.
>
> ## **W1/Q1.** The inference cost is significantly increased. / Comparison to existing CBMs on inference speed.
> Thank you for this comment. We respectfully note that the direct comparison of inference speed between Z-CBMs and existing CBMs is unfair because the problem setting is different, i.e., existing CBMs require training. Even so, in order to identify the limitations of Z-CBMs and the level to which future research should aim, we compared the inference speed. We compared Z-CBMs with Label-free CBMs as follows.
>
> **Table 6-1. Evaluation on ImageNet**
> | Method | Top-1 Accuracy | Inference Time (ms) |
> | :-| :-| :-|
> | Label-free CBM | 58.00| 3.30  |
> | Z-CBM (K=128)  | 54.91| 7.87  |
> | Z-CBM (K=256)  | 57.83| 11.64 |
> | Z-CBM (K=512)  | 60.86| 17.31 |
> | Z-CBM (K=1024) | 61.92| 33.75 |
> | Z-CBM (K=2048) | 62.70| 55.63 |
>
> We can see that ZCBMs are at least twice as expensive to infer as Label-free CBMs. As the concepts contained in images are not known in the zero-shot problem setting, a relatively large value of $K$ to accommodate any concept for high accuracy needs to be selected. In this regard, future work could include approaches such as dynamically creating a cache of concepts according to task. We will add this discussion to the paper.
>
> ## **W2.** On the relevance to “Visual Classification via Description from Large Language Models”
> Thank you for sharing related work. The mentioned paper proposed a zero-shot classification method by LLM-generated text descriptions for each target class. In contrast to Z-CBMs, which directly decompose input features by concepts, this method makes a prediction based on the correlation between the input features and the task-specialized texts. In this regard, this method can also provide interpretability in a different way from Z-CBMs, but it requires (i) generating the task-specialized texts with LLM, (ii) restricting the inference algorithm to the CLIP style zero-shot classification, and (iii) calculating the contribution scores for each text independently. On the other hand, Z-CBMs can be used for arbitrary domains without external LLMs and arbitrary inference algorithms (e.g., training heads). Further, Z-CBMs can provide relative contribution scores among concepts by sparse regression, i.e., we can compare different concepts quantitatively. We will add the discussion to Sec. 5.
>
> ## **Q2.** I’m wondering whether the visual and textual features are in the same space as shown in Fig 2 (a)
> Thank you for this question. We show the PCA feature visualization in Fig.7 of the revised paper; we use PCA instead of t-SNE because PCA provides more realistic intuition in the feature space with computing only linear transformations. We see that, in reality, the clusters of visual and text features are separated. This known as the modality gap, is one of the challenges of CLIP representations [b]. Although mitigating the modality gap is out of the scope of our paper, Z-CBMs can alleviate the modality gap by interpreting input images by textual concepts, as seen in Fig. 7. We consider this is the reason why Z-CBMs improve the zero-shot CLIP in Table 1.
>
> [b] Liang, Victor Weixin, et al. "Mind the gap: Understanding the modality gap in multi-modal contrastive representation learning." NeurIPS 2023.
>
> ## **Q3.** Concepts such as “Not maltese dog terrier” cannot provide interpretable information for identifying categories
> Thank you for this comment. Strictly speaking, we consider that the appropriate granularity of concepts depends on use cases and is undecidable in general. For example, if the user wants a fine-grained classification of dogs, "Not maltese dog terrier" might be useful information for assessing the decision boundary. If one wants to avoid concepts similar to target class names, one can control the appearance by modifying the threshold in the concept filtering (L225). Similarly, we can avoid specific sets of concepts by directly removing them from the concept bank. However, automatically controlling concepts and their granularity in general cases is an open question and should be resolved in future work.

---

> > ### Comment · Reviewer_mgBg · 2024-11-25
> >
> > Thanks for the provided responses. I believe this paper demonstrates innovation and value, but it also has certain shortcomings, as mentioned by myself and other reviewers. For me, the main concern lies in the insufficient interpretability of some concepts used by the authors. Therefore, I will maintain the current score.

---

> > > ### Author Response · Authors · 2024-11-25
> > >
> > > Thank you for reading our rebuttal and providing additional comments. We are delighted that you have found innovation and value in our research. As for the interpretability given by Z-CBM, we acknowledge that the control of the granularity of concepts is not perfect, especially as mentioned in the rebuttal, but we believe that this issue should be advanced in future research as another research question. Nevertheless, we would be honored to discuss this perspective with you. We thank you for your professional review comments, constructive discussions, and timely and knowledgeable replies.

---

### Official Review · Reviewer_Q8YW · 2024-10-28

**Soundness:** 4
**Presentation:** 3
**Contribution:** 1
**Rating:** 3
**Confidence:** 4

**Summary:**

In this paper zero-shot concept bottleneck models are discussed as a means of obtaining explainable (in terms of concepts) zero-shot classifiers (as in classes without explicit training data). The idea is to use a concept bottleneck model to translate an image into a set of visual concepts, and then use these concepts to classify the image into one of the target classes of the benchmark dataset. Since the train/test data of the benchmark data is not explicitly used, this is a form of zero-shot classification. In the proposed method, the concept bank consists of about 5M concepts obtained from image captioning datasets (including e.g. Flickr-30K and the YFCC-15M dataset). The image and all the concepts are encoded in the CLIP embedding space, and then the top K most similar ones are used. From this set of concepts a sparsely weighted CLIP feature vector is constructed, which is then used to find the nearest target class y. This model is evaluated on 12 classification tasks and performs similar to zero-shot CLIP.

**Strengths:**

The ideas presented in this paper make a lot of sense, and the manuscript is clearly written. The relevance becomes clear from the amount of related research in this direction of ‘attribute-based’ or ‘concept-based’ zero-shot classification, which dates back at least to 2010s. However, this is also the largest weak point of the paper, the novelty compared to papers and ideas presented back then is not clearly stated, nor is the paper compared to any other zero-shot method besides retrieving in the CLIP space. Of course some techniques / methods did not exist back then (eg the CLIP embedding space), that does not make this paper substantially different.

**Weaknesses:**

## Major weakness
The major weakness of this submission is the novelty with respect to previous work (likely of a previous generation, before deep learning took off). The idea of zero-shot classification in a visual-semantical space based on a joint embedding is not novel. A good example is [**ConSe 2014**], where imagenet classifiers are used together with a Word2Vec space to compose a Word2Vec embedding for an image (based on the classifier outputs and the word embeddings of the class names], which is then used for zero-shot classification in text space. This is extremely similar to the posed idea, except that now a CLIP space is used. Also the idea of using a (sparse) regression of the concepts has been explored before [**Write 2013, Costa 2014, Objects2Actions 2015**]. None of these papers uses an explicit attribute/concept-to-class mapping as the seminal work of Lampert et al. [**AwA 2013**], they all used a discovered attribute-to-class mapping based on an embedding space [**ConSe 2014, Objects2Actions 2015**] or based on co-occurrence statistics or web search [**Costa 2014**], including co-occurences from the YFCC dataset, also used in this work.

The *only* difference I see with respect to these works, is that the concept bank used in this paper is much larger and that a CLIP embedding is used. Based on the previous works, the following questions are interesting, but not explored in this submission:
- The weighting of concepts is now based on the input image, it could also be done based on the target classes (ie, each class selects the top-K concepts which are most similar, or find the most co-occurring concepts in the captioning datasets)
- The weights of a concept in the linear regression model can be negative, this is unlikely to be beneficial given that the used concepts are the top-K most relevant for this particular image. Would it make sense to restrict W to be positive?
- What is the influence of lambda on the performance? And on the sparsity? Is the optimal lambda dataset specific? It seems that the current value (1x10-5) is extremely small, compared to the size of W (which has K weights, with K ~1000).
- Using proper negative concepts for a class is likely to be beneficial, given that knowing what is not related to the target class is a strong signal, could that be explored as well?
- What similarity function is used in the clip space? Is it cosine similarity? Is Fcx W normalized?
- The similarity between a concept and the image is now an indicator function only (concept in top-K concepts for this image). While, the similarity value might contain a strong signal of relevance. It could make sense to use the similarity value between the image and the concepts also in constructing the concept clip embedding of the image.

## Secondary weaknesses / suggestions
1. The second step, the final label prediction (Eq 4) is a purely textual reasoning problem. In the light of the enormous reasoning power of the LLMs, it could be explored if LLMs would be able to reason about the final class provided the top-K concepts from the previous stage.

2. A suggestion for an additional exploration. In this submission, the CLIP space is searched in a cross-modal setting, from an input image to a target/output text. While in [**ReCo 2024**] it has been shown that uni-modal search works much better (image-image) and then use cross-modal fusion (use the textual description of that image). This could be exploited (e.g.) by using (image, caption) pairs from the image datasets. It would be interesting to study if different search strategies improve the zero-shot classification performance.

### Minor/Nitpicks
- In table 1: the bold facing of performance should include the zero-shot/linear-probe CLIP.
- It is unclear why the zero-shot CLIP model should be considered as the upper bound of the proposed method. The proposed method uses the (implicit) knowledge of millions of additional (image, text) pairs.

# Post Rebuttal Evaluation:
It seems that we reached consensus that this paper brings hardly any technical novelty, but uses existing ideas from zero-shot classification for the zero-shot concept bottleneck model idea. In that light, I believe that the paper should be rewritten completely, putting a fair treatment of related work from the zero-shot classification community, discussing the similarities and the differences, explaining why known methods from these days can't be used, or even better just evaluate at least a handful of these ideas within this new domain. I still don't see the major conceptual difference between describing an action based on imagenet classes (as eg in [*Objects2Actions 2015*]) with describing an class with concepts from a concept bank. Such a rewrite is imho beyond a conference submission revision, hence looking forward seeing a revised version of this paper in the future.



### References

- [**AwA 2013**]: Attribute-based classification for zero-shot visual object categorization, TPAMI 2013.
- [**ConSe 2014**]: Zero-Shot Learning by Convex Combination of Semantic Embeddings, ICLR 2014.
- [**Costa 2014**]: COSTA: Co-Occurrence Statistics for Zero-Shot Classification, CVPR 2014.
- [**Objects2Actions 2015**]: Objects2action: Classifying and localizing actions without any video example, ICCV 2015.
- [**Write 2013**]: Write a Classifier: Zero-Shot Learning Using Purely Textual Descriptions, ICCV 2013.
- [**ReCo 2024**]: Retrieval Enhanced Contrastive Vision-Text Models, ICLR 2024.

**Questions:**

1. The main question is how is this work different from [**ConSe 2014**] (and other similar works), and then beyond that they use an ImageNet classifier to transform images to text, and here a CLIP space has been used. So, please clarify what novel contributions the method makes beyond using a CLIP embedding space instead of Word2Vec + ImageNet classes?

2. Please clarify: (a) the used CLIP similarity function, (b) Fcx W being normalized, (c) the influence of lambda.

3. Please discuss the open directions (taken from previous research): weighing based on target classes, restricting W to positive weights only, using negative concepts in a proper manner, using similarity value.

3. From Figure 3 it becomes clear that some concepts are negated, for example `NOT macro rope` (bottom row, right). How is this defined? Is the `not` a part of the concept, and hence used encoded in `f_T(concept)` vector, or is the `not` a result of the linear regression, for these concepts with a negative weight in W? Please elaborate whether it is conceptually desired that concepts in the top K most related concepts for an image could be negative weighed for the image-text embedding.

---

> ### Author Response · Authors · 2024-11-19
> **Author Response for Q8YW (1/2)**
>
> Thank you for your knowledgeable review and many interesting suggestions. We address your concerns below. We are sorry for the multiple responses but hope that these responses help you to re-evaluate our work.
>
> **Table 5-1. Additional evaluation on ImageNet**
> | Method | Top-1 Accuracy |
> | :- | :- |
> | Zero-shot CLIP | 61.88|
> | Z-CBM (Ours) | **62.70**|
> | ConSe | 25.19|
> | class-to-concept | 61.86|
> | Z-CBM w/ positive weight | 39.77 |
> | uni-modal search & cross-modal fusion | 10.11|
>
> ## **MW1/Q1.** Novelty is limited. How is this work different from [ConSe 2014]?
> Thank you for providing related work. Although they are important and should be discussed, our work is totally different from theirs and has remarkable novelty. Here, we focus on the difference from [ConSe 2014], which is your primary concern.
> First of all, **the zero-shot classification with [ConSe 2014] is completely different from that with Z-CBMs**. ConSe infers a target label from a semantic embedding composed of a weighted sum of concepts of the single predicted ImageNet label, whereas Z-CBMs infer the label by the concept features retrieved from the concept bank and weighted by sparse regression. ConSe has three potential risks: (i) Since the zero-shot inference depends on the ImageNet label space, it cannot accurately predict target labels if there are no target-related labels in ImageNet, (ii) the prediction accuracy largely depends on the model's ImageNet accuracy, (iii) the cosine-similarity based prediction can produce concepts semantically overlapped with each other and thus the accuracy and interpretability is restricted.
> In contrast, our Z-CBMs directly decompose an input image feature into concepts via concept bank, so it is not restricted to any external fixed label spaces. Our concept regression can also provide semantically identical concepts via sparse regression algorithm.
> Furthermore, we compared the performance between ConSe and Z-CBMs; we implemented ConSe with pre-trained CLIP and concept bank, which were in the same setting as Z-CBMs. The results are shown in Table 5-1 and Table 1 of the revised paper. **Our Z-CBMs largely outperformed ConSe, indicating that the methodology of Z-CBMs is superior to and different from ConSe**. We will add ConSe as a zero-shot baseline and discuss it in the main paper.
>
> The other works partially share some technical components with our work (e.g., sparse regression), but they differ greatly from our Z-CBMs: [Costa 2014] also depends on the existing classifier in the same fashion as ConSe, [Write 2013] requires additional training for seen classes, and [Object2Actions 2015] limits the domains to action recognition. More importantly, none of them focus on the interpretability of the models' decisions.
> In contrast, our work is the first work to build interpretable zero-shot concept bottleneck models from the combination of concept retrieval and concept regression using a large-scale concept bank without any additional training and restrictions of domains, as acknowledged the technical novelty by Reviewers xHwx and mgBg.
>
> ## **MW2/Q3.** Weighting concept based on target classes
> Thank you for this suggestion. We did not consider this direction because it does not achieve the goal of concept bottleneck models, i.e., predicting concepts from input, and then predicting final labels from the concepts. This goal is essential to provide the interpretability of the output. Even so, we evaluated this variant as shown in Table 5-1 (the row of "class-to-concept"). The variant of "class-to-concept" slightly decreased the zero-shot baseline performance. This may be because retrieving and weighting concepts from target class texts are not helpful in reducing the modality gap between image and text in contrast to Z-CBMs. Therefore, this direction does not seem promising in terms of either interpretability or performance.
>
> ## **MW3/Q3.** Positive weight constraint on the linear regression would be beneficial
> Since $F_{C_x}$ is defined in the real number space, the retrieved concept vectors also contain negative values. In such a sense, the linear regression requires the negative values of $W$ to reconstruct image features $f_\mathrm{V}(x)$ that are also defined in the real number space. In fact, when constraining the weights to be positive, the performance was significantly degraded ("Z-CBM w/ positive weight" in Table 5-1). Further, the negative concepts are helpful for a more detailed understanding of the decision boundary.

---

> > ### Author Response · Authors · 2024-11-19
> > **Author Response for Q8YW (2/2)**
> >
> > ## **MW4/Q2.** What is the influence of lambda on the performance? Is it dataset-specific?
> > We selected $1.0\times10^{-3}$ as $\lambda$ by searching from $\{1.0\times10^{-2},1.0\times10^{-3},1.0\times10^{-4},1.0\times10^{-5},1.0\times10^{-6},1.0\times10^{-7},1.0\times10^{-8}\}$ to choose the minimum value achieving over 10\% non-zero concept ration when using $K=2048$ on the subset of ImageNet training set. We used the same $\lambda$ for all experiments. We will add this description to Sec. 4.1. We also show the effects of $\lambda$ in Figure 8 in the revised paper. Using different lambda varies the sparsity in concept regression and accuracy. Therefore, selecting appropriate $\lambda$ is important for achieving both high sparsity and high accuracy.
> >
> > ## **MW5/Q3/Q4.** Is using proper negative concepts beneficial? What are NOT concepts in Fig. 3?
> > Yes, Z-CBMs originally used negative concepts, and they are beneficial. The NOT concepts in Fig.3 are defined as the predicted concepts with negative weights by following Oikarinen et al. (2023). In Fig.3, we printed the top 5 concepts when sorting the absolute values of weights. Therefore, "NOT macro rope" can mean that it is not likely "macro rope," providing information on the classification boundaries of the model.
> >
> > ## **MW6/Q2.** What similarity function is used in the clip space? Is it cosine similarity? Is $F_{C_x}W$ normalized?
> > Yes, the similarity function $\mathrm{Sim}(\cdot)$ in Eq. (1) is cosine similarity (L153). $F_{C_x}W$ is not explicitly normalized because $f_\mathrm{V}(x)$ and each concept feature of $F_{C_x}$ in Eq. (3) is normalized, i.e, concept regression finds $W$ to approximate the normalized features in Eq. (3). We will add these descriptions.
> >
> > ## **MW7/Q3.** Is using (cosine) similarity between images and concepts helpful for input-to-concept prediction?
> > Yes, Z-CBMs actually use the cosine similarity in concept retrieval in Eq. (1). Similarly, we also tried to use the cosine similarity for the concept-to-label prediction (concept regression) in Table 6 ("CLIP Similarity"). However, it failed to achieve practical performance, indicating the signals from the cosine similarity are not perfect to represent the concept's importance. This is the reason why we introduce concept regression into the concept-to-label prediction to accurately estimate the importance.
> >
> > ## **SW1.** Do LLMs predict the final labels from top-K retrieved concepts?
> > Technically, yes. However, using LLMs in the prediction completely loses interpretability because LLMs are inherently black-box models constructed from transformers. Since our goal is to build an interpretable model, in such cases, we require another unobvious interpretable method for LLMs to interpret their outputs. In this sense, our Z-CBMs can straightforwardly provide the interpretability represented by concept weights from the sparse regression.
> >
> > ## **SW2.** On the effectiveness of uni-modal search and cross-modal fusion
> > Thank you for the suggestion. The direction of using uni-modal search and cross-modal fusion like [ReCo 2024] is interesting. We implemented this approach by using the union caption dataset of CC3M, CC12M, and YFCC15M. Table 5-1 shows the result. The uni-modal search & cross-modal fusion significantly degraded the performance. This is due to the mismatch between retrieved captions and class label texts. Further, this approach loses direct interpretations of input image features by swapping them to retrieved images' captions.
> >
> > ## In table 1: the bold facing of performance should include the zero-shot/linear-probe CLIP.
> > We will fix the presentation by following your advice.
> >
> > ## It is unclear why the zero-shot CLIP model should be considered as the upper bound.
> > This is because the objective of concept regression defined by Eq. (3) is to approximate the input image features by the concept candidate features. We will revise L283 as "it is expected to perform as the upper bound of the Z-CBMs' performance because Eq. (3) aims to approximate the input image features by concept candidates.

---

### Official Review · Reviewer_xHwx · 2024-11-02

**Soundness:** 2
**Presentation:** 3
**Contribution:** 3
**Rating:** 5
**Confidence:** 2

**Summary:**

This paper utilizes a large-scale concept bank and a dynamic concept retrieval method to make high-accuracy predictions without requiring additional training on a target dataset. By employing sparse regression to identify and weigh the importance of relevant concepts from the bank, Z-CBMs achieve effective concept-to-label mappings while ensuring model interpretability, addressing the limitations of previous concept bottleneck models that relied on extensive manual data collection and training.

**Strengths:**

1. To the best of my knowledge, this paper is the first to propose a zero-shot Concept Bottleneck Model (CBM), marking a significant contribution to the field of CBMs. Furthermore, the proposed zero-shot CBM method exhibits predictive capabilities comparable to those of CLIP, while its architecture enhances the model's interpretability.

2. The experiments presented in this paper are comprehensive and well-executed, encompassing 12 datasets. Despite the absence of suitable benchmarks, the authors have effectively compared their method with zero-shot CLIP and other training head approaches.

3. This paper introduces the concept of a "concept bank" and employs an efficient concept retrieval method for label prediction based on this foundation. The concept bank is constructed through the analysis of extensive datasets. In Section 4.6.2 and Table 1, the authors provide a detailed comparison of zero-shot performance across different sizes of concept banks, demonstrating that expanding the concept bank enhances the expressive capacity of the CBM, thereby improving its zero-shot performance.

**Weaknesses:**

1. While this article provides a valuable comparison of various methods related to the concept bank, it appears that the testing results for a specific approach—constructing a concept bank using a question-and-answer method similar to the label-free CBM [1]—are not included. Including this method, particularly in the context of designing a smaller, domain-specific concept bank, could enhance the comprehensiveness of the analysis. I encourage the authors to include a comparison with a concept bank generated using the question-and-answer approach from the label-free CBM, as this would provide a deeper understanding of the different concept bank construction approaches.

2. The paper mentions that the regular term in sparse regression can help reduce conceptual redundancy; however, it lacks specific visual results to illustrate this effect. Additionally, the advantages of using sparse regression in comparison to other distance metrics in feature space for weight determination are not clearly established. To strengthen the paper, I suggest that the authors provide visual examples comparing the concepts selected by sparse regression versus other methods, demonstrating how redundancy is reduced. Furthermore, including a quantitative comparison of sparse regression against other weighting methods would enhance the clarity and convincing nature of the proposed method.


Reference:
[1]. Oikarinen, Tuomas, et al. "Label-free concept bottleneck models." arXiv preprint arXiv:2304.06129 (2023).

**Questions:**

1. This article compares various methods related to the concept bank. However, I may have overlooked the testing results for a specific approach: constructing a concept bank using a question-and-answer method similar to the label-free CBM [1]. This involves designing a smaller concept bank tailored to the problem domain.

2. In this paper, it is mentioned that the regular term in sparse regression can help reduce conceptual redundancy. Could you please provide some specific visual results to illustrate this effect? Additionally, I’m curious about the advantages of sparse regression compared to using distance or other metrics in feature space to determine weights. If there are any experimental results that demonstrate this comparison, it would certainly enhance the persuasiveness of the method presented in your paper.

3. I noticed the inference time presented in Figure 6. Could the authors clarify whether this represents the total time for the entire zero-shot inference process? As the scale of the concept bank expands, it is important to understand how embedding and concept retrieval times may increase. I would appreciate it if the authors could provide a breakdown of the reported times, detailing the components of the inference process (e.g., embedding, concept retrieval, regression) and how these times are affected as the concept bank size increases.

---

> ### Author Response · Authors · 2024-11-19
> **Author Response for xHwx**
>
> We appreciate your constructive and helpful comments and suggestions. We address your concerns below. We will revise our paper according to your suggestions.
>
> ## **W1/Q1.** Additional experiments using concept bank with question-and-answer (Q&A) approach
> Thank you for this valuable suggestion. We have shown in the GPT-3 (ImageNet Class) row of Table 5 an experiment using a concept bank (4K concepts) with a Q&A approach of Label-free CBM. In order to deeper understand the importance of the concept bank, we now compare it with a baseline utilizing the same concept bank. The results are as follows.
>
> **Table. 4-1 Evaluation on ImageNet with GPT-3 (ImageNet Class) Concepts**
> || Top-1 Acc. | CLIP-Score |
> | :- | :- | :- |
> | Label-free CBM|58.00|0.7056|
> | CDM |62.52|0.7445|
> | Z-CBM (Zero-shot)|59.18|0.6276|
> | Z-CBM (Training Head)|62.73|0.6276|
>
> **Our Z-CBMs achieved competitive performance even when using such a smaller concept bank.** However, they degraded CLIP-Score. This suggests the importance of the concept bank. In our zero-shot setting, covering sufficient concept knowledge with an abundant vocabulary is important to map concept-to-label without learning. We will add this result and discussion to Sec. 4.6.
>
> ## **W2/Q2.** Qualitative and quantitative concept comparisons of linear regression and lasso
> We also thank this insightful suggestion. For qualitative evaluation, we add the concept visualizations of Z-CBMs with linear regression to Fig. 3 of the revised paper. We can see that **linear regression tends to produce concepts that are related to each other**. In fact, quantitatively, we also found that the averaged inner CLIP-Scores among the top-10 concepts of lasso is significantly lower than that of linear regression (0.6855 in lasso vs. 0.7826 in linear regression), which means that **lasso produces more independent concepts than linear regression**. These results emphasize the advantage of using sparse regression like lasso in concept regression to reduce redundancies of the concepts.
>
> ## **Q3.** Does Fig. 6 represent the total time for a zero-shot inference? Provide a breakdown of the reported times.
> Yes, the y-axis of Fig. 6 represents the total time. A detailed breakdown is as follows.
>
> **Table. 4-2 Execution time for zero-shot inference (milliseconds)**
> | $K$    | total | feature extraction | concept retrieval | concept regression |
> | :- | :- | :- | :- | :- |
> | 128  | 7.87  | 0.12 (1.5%)| 1.00 (12.7%)| 6.63  (84.2%)|
> | 256  | 11.64 | 0.11 (1.0%)| 1.68 (14.5%)| 9.69  (83.2%)|
> | 512  | 17.31 | 0.11 (0.7%)| 1.87 (10.8%)| 15.15 (87.5%)|
> | 1024 | 33.75 | 0.12 (0.4%)| 3.11 (9.2%)| 29.88 (88.5%)|
> | 2048 | 55.63 | 0.11 (0.2%)| 5.35 (9.6%)| 49.23 (88.5%)|
>
> For any $K$, the computation time of concept regression is dominant for the total. This is a limitation of Z-CBMs: there is a trade-off between computation time and the accuracy of the zero-shot inference. Therefore, we will address resolving this trade-off and speeding up the zero-shot inference in future work. We will revise 4.6.3 to add this discussion.

---

> > ### Author Response · Authors · 2024-11-25
> >
> > Dear Reviewer xHwx,
> >
> > Thank you again for your constructive and helpful review comments. We respectfully remind you that **the discussion period will end in a few days**. We have responded above to your concerns. We believe these address your concerns. We would appreciate it if you would take the time to read and comment on the responses.
> >
> > Best,
> > Authors

---

> > > ### Comment · Reviewer_xHwx · 2024-11-26
> > >
> > > Dear Authors,
> > >
> > > Thank you for your detailed response. Your experiments have eliminated my concerns.
> > >
> > > I have noted the concerns previously raised by Reviewers Q8YW and FScg. I also doubt the novelty of this paper.
> > >
> > > Therefore, I decided to retain my previous score.
> > >
> > >
> > > Best,\
> > > Reviewer

---

> > > > ### Author Response · Authors · 2024-11-26
> > > >
> > > > Dear Reviewer xHwx,
> > > >
> > > > Thank you for reading our rebuttal. We are glad that our responses have addressed your concerns.
> > > >
> > > > > I have noted the concerns previously raised by Reviewers Q8YW and FScg. I also doubt the novelty of this paper.
> > > >
> > > > For the novelty of our work, we have explained the details based on scientific facts in the responses, including additional evaluations and a discussion of related work in the revised paper. Thus, we would be happy if you could decide the final score by reading the discussion fairly.
> > > >
> > > > Finally, we appreciate you providing your detailed and helpful review comments and taking the time to read our rebuttal.
> > > >
> > > > Best,
> > > >
> > > > Authors

---

> > > > > ### Author Response · Authors · 2024-11-29
> > > > >
> > > > > Dear Reviewer xHwx,
> > > > >
> > > > > Thank you for participating in the discussion. We have addressed your concern in the general response (Clarification on the novelty of our work). We respectfully request you to consider raising your rating score accordingly if your concerns are alleviated. Otherwise, we would be happy to hear the remaining concerns that prevent you from doing so and continue to discuss them.
> > > > >
> > > > > Best,
> > > > >
> > > > > Authors

---

> > > > > > ### Author Response · Authors · 2024-12-01
> > > > > > **Reminder of end of discussion period**
> > > > > >
> > > > > > Dear Reviewer xHwx,
> > > > > >
> > > > > > Thank you for your effort in this review process. We sincerely remind you that **the extended discussion period will end in a few days**. Since we have addressed your remaining concerns above and agreed with other reviewers on the novelty of our works, we would be happy if you could read them and update your score or leave additional comments. We are sure you, the knowledgeable reviewer, will re-evaluate our paper based on them. Finally, we deeply appreciate your participation in this long discussion and would like your thoughts.
> > > > > >
> > > > > > Best,
> > > > > >
> > > > > > Authors

---

### Official Review · Reviewer_FScg · 2024-11-02

**Soundness:** 3
**Presentation:** 3
**Contribution:** 3
**Rating:** 5
**Confidence:** 4

**Summary:**

The authors propose a variant of concept bottleneck models (CBM) which uses sparse linear regression on a databank of concepts to approximate the visual feature of each image. The resulting CBM can achieve reasonable zero-shot accuracy and CLIP-score without additional training. The proposed framework does not require additional data.

**Strengths:**

(1) The method is simple.

(2) The results are good. The authors show that their ZS-CBM achieves SoTA accuracy among prior CBMs. They also demonstrate the quality (relevance) of the selected concepts using CLIP-score results.

**Weaknesses:**

I can't find anything wrong with this paper except perhaps the lack of technical innovation. There is abundant literature on concept bottleneck models. Sparse regression on concept features is very widely used. Using retrieval to find relevant concepts is not technically interesting. In my opinion, this work does not add much value to the existing CBM literature.

**Questions:**

None.

---

> ### Author Response · Authors · 2024-11-19
> **Author Response for FScg**
>
> Thank you for your positive review.
>
> > I can't find anything wrong with this paper except perhaps the lack of technical innovation. There is abundant literature on concept bottleneck models. Sparse regression on concept features is very widely used. Using retrieval to find relevant concepts is not technically interesting. In my opinion, this work does not add much value to the existing CBM literature
>
> Thank you for mentioning this. We respectfully note that our paper has a technical innovation in addition to the existing CBMs. First of all, the idea of building CBMs for arbitrary vision-language models in a zero-shot manner is technically novel, as acknowledged by Reviewers xHwx and mgBg. Specifically, the design of Z-CBMs that search input-related concept candidates from large-scale concept banks and then predict the importance of each concept by sparse regression is not obvious and has significant novelty. Also, since this paper is the first work on zero-shot CBMs, it is important to solve the problem as simply as possible as a baseline for future work. To this end, designing a method with well-known technical components such as retrieval and sparse regression is reasonable. As you commented, simplicity is also a strength, so we would be happy if you look at the contributions that have made CBM possible with zero-shots, rather than the simplicity of individual techniques.

---

> > ### Author Response · Authors · 2024-11-25
> >
> > Dear Reviewer FScg,
> >
> > Thank you again for your positive and constructive review comments. We respectfully remind you that the discussion period will end in a few days. We have responded above to your concerns about the technical innovation of our paper. We would appreciate it if you would take the time to read and comment on the responses.
> >
> > Best,
> > Authors

---

> > > ### Comment · Reviewer_FScg · 2024-11-25
> > >
> > > Thank you for the response. I strongly concur with Reviewer Q8YW that the novelty in this paper is limited, and the response here does not alleviate this concern from my point of view.
> > >
> > > > Specifically, the design of Z-CBMs that search input-related concept candidates from large-scale concept banks and then predict the importance of each concept by sparse regression is not obvious and has significant novelty. Also, since this paper is the first work on zero-shot CBMs, it is important to solve the problem as simply as possible as a baseline for future work.
> > >
> > > I agree that the specific combination of methods is novel. However, the idea of "large-scale concept banks" is not new. Searching for this in Google Scholar yields related papers, such as this one: https://arxiv.org/pdf/1403.7591 from 2014. "predicting the importance of each concept by sparse regression" is certainly not new. LIME uses this.
> > >
> > > I get that the authors are positioning their work as a novel problem setting and establishing a simple and intuitive baseline. Therefore, I maintain my original positive score.

---

> > > > ### Author Response · Authors · 2024-11-26
> > > >
> > > > Thank you for reading our rebuttal and providing the clarification.
> > > >
> > > > > I agree that the specific combination of methods is novel.
> > > >
> > > > > I get that the authors are positioning their work as a novel problem setting and establishing a simple and intuitive baseline.
> > > >
> > > > We are glad to see that you recognize our paper's claim and make some concessions about the novelty of our work in the context of interpretable CBMs.
> > > >
> > > > > However, the idea of "large-scale concept banks" is not new. Searching for this in Google Scholar yields related papers, such as this one: https://arxiv.org/pdf/1403.7591 from 2014.
> > > >
> > > > Thank you for providing related work. At a high level, we agree that this work and several existing works share the idea of using a concept bank. Nevertheless, it is worth noting that our Z-CBMs, unlike these works, enable the use of a concept bank with millions of concepts and the application of models to broad domains in a zero-shot manner. We are aware that you may know such a difference, but we would appreciate your consideration in this regard for the final evaluation.
> > > >
> > > > Finally, we deeply appreciate you providing positive and knowledgeable review comments and insightful discussions.

---

### Official Review · Reviewer_Rb4A · 2024-11-03

**Soundness:** 3
**Presentation:** 3
**Contribution:** 2
**Rating:** 6
**Confidence:** 5

**Summary:**

The paper proposes a “zero-shot” concept bottleneck model (CBM). The original idea in CBM is to first represent a given image in the concept space, ie. as a weighted combination of existing concepts and then classifying the image using these concept-weights based image representation. The original works rely supervised training to build image→concept and concept→class predicts. More recent works reduce labelling efforts by leveraging the prior knowledge available in vision-language models (VLMs) and LLMs, e.g. “Label-Free Concept Bottleneck” and others.

The recent works on avoiding supervised concept predictor training however still require a training dataset to train concept→class predictor. This paper basically aims to avoid supervision altogether. The proposed method semi-automatically builds a large concept vocabulary (over existing image caption datasets) , selects the most relevant concepts specifically for a given image according to the image-text representations of a pretrained VLM (eg CLIP) and learns an image-specific concept-text-embeddings to image-embeddings mapping based on reconstruction loss. The resulting concept-embedding based approximation to image’s representation is used to classify the image based on cosine similarity to the textual embeddings of target class labels.


# Post-discussion update
I would like to thank the authors and the fellow reviewers for the fruitful discussions. I’ve read the discussions (including the final responses of the authors), and here is a summary of my opinion:

- I expressed my concerns about the possibly misleading results in Table 1, and the random matrix experiment confirms these concerns. The revised paper, however, mitigates this issue by providing a more cautious discussion and supporting it with additional results. I believe the random matrix experiment should also go into any published version of the paper.
- I found the CLIP-Scores misleading and authors agreed on that. They proposed to address this concern by separating the CLIP model used within the method from the one used in evaluation. While the authors' proposal to separate the CLIP model used in evaluation reduces some bias, it is likely that CLIP models, despite low-level differences, behave in correlated ways. This limitation makes it challenging to establish CLIP-Score as a fully reliable metric.
- I’m “happy” about the hyper-parameter tuning response of the authors, thanks.
- Literature discussion is improved but here I do share the concerns of Reviewer mgBg: the arguments remain too strong in many places, like claiming the model as free of from additional training or data. The fact that somebody else trained CLIP on a gigantic dataset doesn’t alter the dependencies of the proposed approach. The paper shares elements with prior works built on pre-trained classifiers and visual attributes more than what the current text reflects, even after the updates.
- Domain dependence: this is still an open concern, but perhaps not a red flag on its own.

Overall, I remain inclined towards a ‘weak reject’ due to the concerns outlined above. However, in recognition of the thoughtful improvements addressing some of these concerns, I am raising my score to 6.

**Strengths:**

- The paper’s work is an interesting addition to the research on CBMs. It addresses the missing supervision problem that seems to remain unaddressed in prior CBM work and tackles it in a relatively meaningful manner.
- The XAI-performance results in Table 3 look impressively good.
- The method is simple and easy to understand.

**Weaknesses:**

- The paper’s results in Table 1 are not impressive , and this is very much expected as the learned CBM-based representation is afterall an image-specific approximation to the image’s visual representation. It is “normal” that it performs very similar to the CLIP baseline, and I am not sure if the improved performance implies any significant achievement, as the paper lacks any substantial analysis on it (despite commenting that it might be thanks to the reduced representation gap).
- It seems to me that the paper’s results in Table 2 (CLIP-Score) can be misleading because it seems to be measuring the average correlation between the image’s CLIP-image representation and the obtained CBM representation. As the CBM representation of this paper is a direct reconstruction of CLIP-image representation, it seems again “normal” (not interesting) to observe high scores. (Please correct me if I’m missing something here.)
- How were the hyper-parameters like lambda tuned? Is lambda (and other hyper-parameters, if exists) all the same across all experiments and all datasets? What methodology was used? ie. if one wants to reproduce the exact results from scratch, how should he/she tune the hyper-parameter(s) to reach the same value(s).
- The paper seems to be missing one relevant paper from the zero-shot learning domain: “Attributes2Classname: A discriminative model for attribute-based unsupervised zero-shot learning”. Similar to the proposed work, this paper learns to represent images in terms of a linear transformation of relevant concepts’ (predicted attributes’) textual embeddings, with and without labelled image dataset. It seems to share many motivations like reducing modality gap via representing images in terms of a combination of concept (attribute) textual embeddings and avoiding image supervision, and therefore can/should be discussed within the paper.
- The method heavily relies on the prior knowledge of pre-trained VLM (CLIP), and therefore, cannot be used in incompatible domains; unlike (more) supervised CBMs. In that sense, as this paper already relies on a huge training set that the VLM pre-training requires, it is not clear if any real achievement is made in terms of building human-understandable concept-based image representations with reduced supervision, from a philosophical point of view.

**Questions:**

- Can you provide more detailed comparisons to prior work by directly using their concept sets? I am a bit lost in understanding how much the comparisons to prior work are directly one-to-one comparable and what elements make a (positive/negative) difference?
- How does the hyper-parameters like lambda for linear regression and lasso affect the results in terms of Table 1, 2, 3 results?
- In regards to the performance gains over CLIP image embedding: how well the method performs if you were to use a random matrix as $F_{C_x}$ in Eq 3 & 4, instead of true concept embeddings, for various K values?

**Details Of Ethics Concerns:**

-

---

> ### Author Response · Authors · 2024-11-19
> **Author Response for Rb4A**
>
> We appreciate your detailed and professional review and address your concerns below. We will revise our paper accordingly.
>
> ## **W1.** Accuracy is not impressive, and the paper lacks analysis of the modality gap.
> Thank you for this comment. We respectfully remark that our main contribution is building practical zero-shot interpretable models without target datasets or training (L078). Table 1 supports our practicality claim, as naive methods struggle to achieve competitive performance (Table 6). To analyze the modality gap, we followed [b] and measured L2 distances: $1.74\times 10^{-3}$ for image-to-label and $0.86\times 10^{-3}$ for concept-to-label features. This demonstrates that **Z-CBMs significantly reduce the modality gap via concept regression**. Additionally, Fig. 7 shows PCA feature visualizations, indicating that weighted concept sums effectively bridge image and text modalities. These findings highlight that our results are not "normal" and offer novel insights.
>
> [b] Liang, Victor Weixin, et al. "Mind the gap: Understanding the modality gap in multi-modal contrastive representation learning." NeurIPS 2023.
>
> ## **W2.** CLIP-Score may be misleading since it uses reconstructions from image features.
> Apologies for any confusion. Table 2 presents averaged scores of the top-10 concepts, selected by sorting absolute regression coefficients without using reconstructed vectors from Eq. (3). We also did not weigh concepts by coefficients when computing CLIP-Score. Thus, **the scores are not obvious, as they depend on the regression algorithm used**. For example, lasso outperforms linear regression, indicating better concept selection (Table 2-1).
>
> **Table 2-1. Evaluations of Z-CBMs on ImageNet**
> |Regression Alg.|Top-1 Acc.|CLIP-Score|
> |:-|:-|:-|
> |Linear Regression|52.88|0.7076|
> |Lasso|62.70|0.7746|
>
> ## **W3/Q2.** How was $\lambda$ tuned, and how does it affect results?
> We selected $\lambda = 1.0\times10^{-5}$ by searching from ${10^{-2},10^{-3},\dots,10^{-8}}$, aiming for a non-zero concept ratio above 10\% with $K=2048$ on a subset of ImageNet. This $\lambda$ was used consistently across all experiments. As shown in Fig. 8 of the revised paper, varying $\lambda$ affects concept sparsity and accuracy, underscoring the importance of careful $\lambda$ selection for balancing sparsity and performance.
>
> ## **W4.** On the relevance to "Attributes2Classname (A2C)"
> Thank you for sharing related work. Indeed, A2C shares ideas of predicting textual concepts (attributes) from images and then predicting the final class from the concepts.
> However, unlike A2C, which requires supervised training for both image-to-concept and concept-to-label mappings, our Z-CBMs operate without additional training or datasets.
>
> ## **W5.** Z-CBMs rely on CLIP and cannot be used in incompatible domains
> This is an inherited limitation of Z-CBMs and VLM-based CBMs (e.g., Label-free CBMs). Nevertheless, we already have domain-specific CLIP models like MedCLIP [c] and can easily expect such CLIP variants to appear in each domain. In such a sense, one advantage of Z-CBMs over other CBMs is the availability without learning when such foundation models emerge. Besides, giving interpretability to the foundation models through Z-CBMs will also serve as a baseline for building supervised CBMs, making a fundamental contribution to this area.
>
> [c] Wang, Zifeng, et al. "Medclip: Contrastive learning from unpaired medical images and text." EMNLP 2022.
>
> ## **Q1. Can you provide more detailed comparisons to prior work with their concept sets?**
> Yes. We have already shown the Z-CBM's result of GPT-3 generated concepts of Label-free CBMs in Table 6. Here, we compare the performance when using the identical concept set:
>
> **Table. 2-2 Evaluation on ImageNet with GPT-3 (ImageNet Class) Concepts**
> ||Top-1 Acc.|CLIP-Score|
> |:-|:-|:-|
> |Label-free CBM|58.00|0.7056|
> |CDM |62.52|0.7445|
> |Z-CBM (Zero-shot)|59.18|0.6276|
> |Z-CBM (Training Head)|62.73|0.6276|
>
> Although this is not a fair comparison since the baselines learn concept-to-label mapping on supervised datasets while Z-CBMs do not, our Z-CBMs achieved competitive performance. However, the CLIP-Scores are lower than Z-CBMs (All) in Table 2. This suggests the importance of using a large-scale concept bank to accurately map concept-to-label without learning.
>
> ## **Q3. How well the method performs if using random matrix as concept features $F_{C_x}$?**
> We evaluated this case on ImageNet as follows:
>
> **Table 2-3. Top-1 Accuracy on ImageNet**
> |$K$|Z-CBM|Random|
> | :- | :- | :- |
> |128|54.91|13.90|
> |256|57.83|36.08|
> |512|60.86|50.41|
> |1024|61.92|55.91|
> |2048|62.70|61.88|
>
> Using random matrices gradually approach the zero-shot baseline performance (61.88) as $K$ increases. This is because larger numbers of random vectors have more chance to contain similar vectors to image features. However, **real concepts of Z-CBMs always outperformed random concepts, demonstrating the advantage of using meaningful concepts**.

---

> > ### Author Response · Authors · 2024-11-25
> >
> > Dear Reviewer Rb4A,
> >
> > Thank you again for your detailed and constructive review comments. We respectfully remind you that **the discussion period will end in a few days**. We have responded above to your concerns. We believe these address your concerns. We would appreciate it if you would take the time to read and comment on the responses.
> >
> > Best,
> > Authors

---

> > > ### Comment · Reviewer_Rb4A · 2024-11-26
> > >
> > > Dear Authors,
> > >
> > > Thanks for your careful responses.
> > >
> > > # W1
> > > I think there is a bit of misunderstanding regarding my comment. As opposed to the summary given as part of the author response, I do not say that "Accuracy is not impressive" (which would imply an absolute sense of accuracy), and "the paper lacks analysis of the modality gap" is not really my point. What I mean is the following: we can see $F_{C_x}$ as a set of *image-specific* "basis" vectors, or an image-specific dictionary. If this dictionary is large/rich-enough in terms of the number of vectors and not being highly correlated with each other, then Eq 3 may result in an arbitrarily close approximation to the image's CLIP representation (when the concepts do or do not make sense). Therefore, I would not normally expect an improvement over the CLIP baseline. That's why I had written "I am not sure if the improved performance implies any significant achievement, as the paper lacks any substantial analysis on it (despite commenting that it might be thanks to the reduced representation gap)".
> > >
> > > In fact, the newly added random matrix experiment (Q3) seems to support my concerns: it appears that K=2048 by default in the experiments, and the performance gap between a purely random set of vectors versus CBMs seems to diminish greatly. The proposed method creates an image-specific vocabulary with vectors in the CLIP embedding space, therefore, it seems "not so surprising" to obtain a pretty good approximation to the original image embedding. In this sense, these results do not support the claim that that the provided scheme achieves the desired explainability.
> > >
> > >
> > > # W2 (and Q1)
> > >
> > > I've noticed the new explanation added to the paper. Still, what I don't find surprising is the fact that the concept weights are selected (in Eq 3) to maximize their correlation with the CLIP feature. Therefore, again, it looks not so surprising that the top weighted ones lead to a high CLIP-Score. Please let me know if I'm missing anything.
> > >
> > > # W3/Q2. How was tuned, and how does it affect results?
> > >
> > > Is the subset that you used a part of the training data,  or validation / test split? Which performance metric did you use for hyper-parameter tuning (model selection)?

---

> > > > ### Author Response · Authors · 2024-11-27
> > > >
> > > > Dear Reviewer Rb4A,
> > > >
> > > > Thank you for reading our rebuttal and for providing clarification.
> > > >
> > > > ## **W1**
> > > >
> > > > > we can see $F_{C_x}$ as a set of image-specific "basis" vectors, or an image-specific dictionary. If this dictionary is large/rich-enough in terms of the number of vectors and not being highly correlated with each other, then Eq 3 may result in an arbitrarily close approximation to the image's CLIP representation (when the concepts do or do not make sense). Therefore, I would not normally expect an improvement over the CLIP baseline. That's why I had written "I am not sure if the improved performance implies any significant achievement, as the paper lacks any substantial analysis on it (despite commenting that it might be thanks to the reduced representation gap)".
> > > >
> > > > Sorry for our misunderstanding on your comments. In this regard, we consider that this performance improvement is a side effect that occurred in the process of achieving the objective of the Z-CBMs (i.e., building zero-shot CBMs) and is not really related to the main claim. As you explained above and we mentioned in the submitted paper, the CLIP baseline should be the upper bound in terms of performance. Even so, Figure 7 shows that the modality gap can be reduced by the reconstructed features by Eq. (3), and Table 4 shows that performance improvement occurs when the backbones have relatively weak cross-modal alignment performance. We see that these are interesting findings connected to the existing literature on the modality gap.
> > > >
> > > > > In fact, the newly added random matrix experiment (Q3) seems to support my concerns: it appears that K=2048 by default in the experiments, and the performance gap between a purely random set of vectors versus CBMs seems to diminish greatly. The proposed method creates an image-specific vocabulary with vectors in the CLIP embedding space, therefore, it seems "not so surprising" to obtain a pretty good approximation to the original image embedding. In this sense, these results do not support the claim that the provided scheme achieves the desired explainability.
> > > >
> > > > We agree that the concept regression provides a good approximation of an image embedding if the concept candidates are sufficient. Nevertheless, it is important to note that our aim is not just to approximate image embeddings in any way but to estimate the importance of concept candidates from concept retrieval through approximating image embeddings. In fact, as shown in Table 5, Z-CBMs struggle to approximate the image embeddings when using smaller concept banks even though they use the same $K=2048$, i.e., concept retrieval does not necessarily return sufficient concept vocabulary. This indicates that the final approximation results depend on both concept retrieval and concept regression. In this sense, comparing Z-CBMs and the sparse regression with random vectors may be somewhat misleading because the uniformly sampled random vectors have more of a chance to sufficiently cover the image embeddings, and they are not interpretable.
> > > >
> > > > ## **W2 (and Q1)**
> > > >
> > > > > I've noticed the new explanation added to the paper. Still, what I don't find surprising is the fact that the concept weights are selected (in Eq 3) to maximize their correlation with the CLIP feature. Therefore, again, it looks not so surprising that the top weighted ones lead to a high CLIP-Score. Please let me know if I'm missing anything.
> > > >
> > > > From this series of discussions, we noticed that the root cause of this concern was the use of the same CLIP for CLIP-Score evaluation as for inference. Here, we provide the partial CLIP-Score results with **CLIP-ViT-B/16**, which is not used to implement Z-CBMs in our experiments.
> > > >
> > > > **Table 2-4. CLIP-Score Evaluation on ImageNet**
> > > > |Method|CLIP-Score|
> > > > | :- | :- |
> > > > |Label-free CBM|0.7182|
> > > > |LaBo|0.7341|
> > > > |CDM|0.7629|
> > > > |Z-CBM|0.7848|
> > > >
> > > > These results suggest that Z-CBMs can provide input-related concepts even in another multi-modal embedding space. We will replace all of the results by using CLIP-ViT-B/16. Please let me know if we are missing points.
> > > >
> > > >
> > > > ## **W3/Q2. How was tuned, and how does it affect results?**
> > > > > Is the subset that you used a part of the training data, or validation / test split? Which performance metric did you use for hyper-parameter tuning (model selection)?
> > > >
> > > > Thank you for the question, and sorry for missing information. We searched $\lambda$ on the subset of the training split of ImageNet. We selected a model with the minimum $\lambda$ achieving over 10% non-zero concept ratio when using $K=2048$.
> > > >
> > > > We sincerely appreciate your thoughtful feedback. If there is any misunderstanding, we would appreciate it if you could let us know.
> > > >
> > > > Best,
> > > >
> > > > Authors

---

> > > > > ### Author Response · Authors · 2024-11-29
> > > > >
> > > > > Dear Reviewer Rb4A,
> > > > >
> > > > > Thank you for participating in the discussion. We have addressed your concern in the above. We respectfully request you to consider raising your rating score accordingly if your concerns are alleviated. Otherwise, we would be happy to hear the remaining concerns that prevent you from doing so and continue to discuss them.
> > > > >
> > > > > Best,
> > > > >
> > > > > Authors

---

> > > > > > ### Author Response · Authors · 2024-12-01
> > > > > > **Reminder of end of discussion period**
> > > > > >
> > > > > > Dear Reviewer Rb4A,
> > > > > >
> > > > > > Thank you for your effort in this review process. We sincerely remind you that the extended discussion period will end in a few days. Since we have addressed your remaining concerns above, we would be happy if you could read them and update your score or leave your additional comments. We are sure that you, the knowledgeable reviewer, will re-evaluate our paper based on them. Finally, we deeply appreciate your participation in this long discussion.
> > > > > >
> > > > > > Best,
> > > > > >
> > > > > > Authors

---

### Official Review · Reviewer_ReJr · 2024-11-04

**Soundness:** 3
**Presentation:** 3
**Contribution:** 2
**Rating:** 5
**Confidence:** 3

**Summary:**

This paper addresses zero-shot scenario for concept bottleneck models (CBMs). Previous methods successfully eliminate a dependency of manually annotated concept labels via large language models (LLMs) and vision-language models (VLMs). However, they still require training the models on target dataset and not applicable to the zero-shot scenarios. Zero-shot CBMs (Z-CBMs) constructs large-scale concept bank using caption datasets and noun parser, retrievals concept candidates following input images, and predicts final labels using the retrieved concepts. The experiments demonstrate the effectiveness of the proposed method on target task performance and interpretability.

**Strengths:**

- The paper is well-written and easy-to-follow.
- The main idea is straightforward and intuitive.
- Target task performance is competitive. Table 1 shows that the proposed method even outperforms the performance of the original CLIP, and Table 2 shows that a simple trainable variant of the proposed method outperforms the previous method in the same setting.

**Weaknesses:**

- The reason for performance improvement compared to the original CLIP is unclear. The paper argues that it is due to a reduced modality gap in the concept-to-label mapping. However, this claim is not fair since the modality gap still exists in the input-to-concept mapping. Furthermore, since CLIP is trained on image-to-text matching, the claim that performance improves due to a reduced modality gap in text-to-text matching also requires sufficient references.

- I'm not entirely clear on the advantages of this approach over the most basic interpretable approach based on CLIP. Specifically, one could retain the standard CLIP classification process and simply retrieve concepts from a concept bank using visual features for interpretability. While this baseline is hard to address concept intervention, it doesn't seem to offer significant differences in terms of interpretability.

- The performance difference between linear regression and lasso in Table 6 is unclear. Linear regression should estimate the original visual features ($f_V(x)$) more accurately, so why does linear regression perform so poorly here?

**Questions:**

- Why was linear regression used instead of lasso in L426-427?

---

> ### Author Response · Authors · 2024-11-19
> **Author Response for ReJr**
>
> We appreciate your constructive and insightful feedback. We address your concerns below. We will revise our paper according to your suggestions.
>
> ## **W1.** Why does the proposed method improve the original CLIP? Is the modality gap really reduced?
> Thank you for this valuable comment. Existing research [a] showed that converting the modality improves performance by reducing the modality gap. To further evaluate the modality gap, we compare the distances (modality gap) among images, weighted sums of concepts by Eq.(3), and ground-truth class prompt texts on the feature spaces with ImageNet samples by following [b]. The L2 distances were $1.74\times 10^{-3}$ in image-to-label and $0.86\times 10^{-3}$ in concept-to-label, demonstrating that there is indeed the modality gap, and Z-CBMs largely reduce it by representing images with textual concepts. Fig. 7 in the revised paper also shows the PCA feature visualizations, where the weighted concepts are located between image and text feature clusters, emphasizing the reduction of the modality gap. We will add this discussion with the reference and additional experiments to Sec. B.1.
>
> [a] Qian, Qi et al. "Intra-modal proxy learning for zero-shot visual categorization with clip." NeurIPS 2023.
>
> [b] Liang, Victor Weixin, et al. "Mind the gap: Understanding the modality gap in multi-modal contrastive representation learning." NeurIPS 2023.
>
> ## **W2.** On the advantages of Z-CBMs compared to a simple interpretation of embedding.
> One clear advantage of Z-CBM is that it ensures interpretability by guaranteeing that the final prediction is made from interpretable concepts. That is, **we cannot ensure interpretability by just retrieving concepts because this does not guarantee that the concepts really contribute to the prediction**. To ensure interpretability, our Z-CBMs estimate the importance of the concepts by concept regression and predict the final labels by actually using the weighted concept features. As you commented, intervention is also an advantage of Z-CBMs since it provides meaningful interpretations by changing parts of concepts. Further, the performance improvements by reducing the modality gap can be an advantage, as shown above.
>
> ## **W3.** On the performance difference between linear regression and lasso
> **This is due to the unstable numerical computation of linear regression**. If the feature dimension $d$ is smaller than the concept retrieval size $K$, the Gram matrix of $F_{C_x}$ in linear regression will be rank-deficient, i.e., there is no inverse matrix for the closed-form solution, and lasso can avoid this by sparse regularization. In our setting, we used $d=512$ and $K=2048$ as the default, so the unstable computation prevents the optimization from convergence.  Even if this is not the case, the Gram matrix might be rank-deficient since concept retrieval can generate concepts that correlate with each other. We will add this explanation to Sec. 4.6.4.
>
> ## **Q1.** Why was linear regression used instead of lasso in the intervention experiments?
> **This is because the intervened concepts are already sparse**. In Sec. 4.5, we intervened the concepts with non-zero coefficients after sparse regression. Since the number of non-zero concepts is smaller than the feature dimension, we do not need to consider the unstable computation problem discussed above. If lasso is used here, it is inappropriate as an intervention experiment because the ground-truth concept does not influence the output when sparse regularization eliminates it.

---

> > ### Author Response · Authors · 2024-11-25
> >
> > Dear Reviewer ReJr,
> >
> > Thank you again for your insightful review comments. We respectfully remind you that **the discussion period will end in a few days**. We have responded above to your concerns. We believe these address your concerns, especially on the modality gap reduction. We would appreciate it if you would take the time to read and comment on the responses.
> >
> > Best,
> > Authors

---

> > ### Comment · Reviewer_ReJr · 2024-11-26
> >
> > Dear Authors,
> >
> > Thank you for your detailed response.
> >
> > While the response addressed some of my concerns, several issues remain.
> >
> > First, your explanation that performance improvements over the original CLIP are due to reducing the modality gap is unconvincing. As I mentioned earlier, the modality gap still exists in the image-to-concept matching stage. It’s unclear how introducing an intermediate “concept” stage in Z-CBM, rather than direct image-to-label matching, effectively reduces this gap. Moreover, if reducing the modality gap were truly the key factor behind the performance improvement, Z-CBM should consistently outperform direct image-to-label matching across various backbones, but that is not the case.
> >
> > Second, I have also read the comments from other reviewers and share their concerns regarding the limited novelty.
> >
> > Based on these reasons, I retain my original score.

---

> > > ### Author Response · Authors · 2024-11-26
> > >
> > > Dear Reviewer ReJr,
> > >
> > > Thank you for reading our rebuttal and providing clarifications.
> > >
> > > > First, your explanation that performance improvements over the original CLIP are due to reducing the modality gap is unconvincing. As I mentioned earlier, the modality gap still exists in the image-to-concept matching stage. It's unclear how introducing an intermediate "concept" stage in Z-CBM, rather than direct image-to-label matching, effectively reduces this gap.
> > >
> > > We agree that the modality gap still exists in image-to-concept matching. Conversely, because of the modality gap, the approximation by the textual concept vector of image embeddings by sparse regression is not perfect, i.e., the regression losses are not zero, as we can see in the image and reconstructed feature clusters in Fig 7. Therefore, we consider that such imperfect text-modal approximation of image embeddings results in an improved modality gap during concept-to-label matching, leading to improved performance.
> > >
> > > > Moreover, if reducing the modality gap were truly the key factor behind the performance improvement, Z-CBM should consistently outperform direct image-to-label matching across various backbones, but that is not the case.
> > >
> > > We do not claim that Z-CBMs always improve the zero-shot CLIP baseline in arbitrary cases. Table 4 shows that the performance improvement is caused when the backbone CLIP has relatively small parameters and lower performance. This indicates that Z-CBMs can help the model improve performance when the backbone's modality alignment capability is not so strong. We also respectfully note that performance improvement is not the main contribution of our work. So, the analysis of the modality gap is for explaining why Z-CBMs improve the CLIP baseline in some cases, not for supporting our main claim, i.e., building zero-shot CBMs.
> > >
> > > > Second, I have also read the comments from other reviewers and share their concerns regarding the limited novelty.
> > >
> > > For the novelty of our work, we have explained the details based on scientific facts in the responses, including additional evaluations and a discussion of related work in the revised paper. Thus, we would be happy if you could decide the final score by reading the discussion fairly.
> > >
> > > Best,
> > >
> > > Authors

---

> > > > ### Author Response · Authors · 2024-11-29
> > > >
> > > > Dear Reviewer ReJr,
> > > >
> > > > Thank you for participating in the discussion. We have addressed your concern in the above and general response (Clarification on the novelty of our work). We respectfully request you to consider raising your rating score accordingly if your concerns are alleviated. Otherwise, we would be happy to hear the remaining concerns that prevent you from doing so and continue to discuss them.
> > > >
> > > > Best,
> > > >
> > > > Authors

---

> > > > > ### Author Response · Authors · 2024-12-01
> > > > > **Reminder of end of discussion period**
> > > > >
> > > > > Dear Reviewer ReJr,
> > > > >
> > > > > Thank you for your effort in this review process. We sincerely remind you that **the extended discussion period will end in a few days**. Since we have addressed your remaining concerns above, we would be happy if you could read them and update your score or leave your additional comments. We are sure that you, the knowledgeable reviewer, will re-evaluate our paper based on them. Finally, we deeply appreciate your participation in this long discussion period.
> > > > >
> > > > > Best,
> > > > >
> > > > > Authors

---

### Author Response · Authors · 2024-11-28
**Clarification on the novelty of our work**

Dear Reviewers,

Thank you for reading our rebuttal and participating in the discussion. Since several reviewers question the novelty of our work, let us clarify the position and novelty here.

- We claim novelty in (i) proposing a new zero-shot problem setting of concept bottleneck models (CBMs) where we do not train models with additional training datasets and (ii) building simple yet practical zero-shot concept bottleneck models (Z-CBMs) by combining concept retrieval with large-scale concept banks and sparse concept regression to estimate the importance of the retrieved concepts.
- We do not claim the novelty of each technical component of Z-CBMs, but this does not hurt our novelty of (ii). We acknowledge that several existing studies have already used concept banks to retrieve related concepts and sparse regression to estimate the contributions of input variables. However, to our knowledge, Z-CBM achieves the first successful results to build CBMs in a zero-shot manner. We used these well-studied technical components in Z-CBMs to focus on building a simple baseline for this new problem setting.
- While the individual technical components are not unique, we show the validity of the Z-CBM's design through the ablation studies of concept banks (Table 5) and regression algorithms (Table 6). Additionally, inspired by the comments from Reviewer Q8YW, we also show the comparison results using the existing zero-shot classification baseline (ConSe) in Table 1. These suggest that the combination of concept retrieval and concept regression for building CBMs is unique, and the design is reasonable for achieving practical performance.
- We also agree with Reviewer Q8YW's opinion that we should discuss the similarity between Z-CBMs and existing works from broader perspectives. Thus, we will add more detailed and careful discussions with respect to zero-shot classification to the main paper by extending Section C.

We hope these clarifications eliminate your concerns about the novelty of our work.

Finally, again, our primary contribution is opening up a new research field of zero-shot CBMs, not the novelty of individual technical components.

Best,

Authors

---

### Meta-Review · Area_Chair_QfmU · 2024-12-16

**Metareview:**

This paper tries to handle the zero-shot scenario in concept bottleneck models, which is different from the existing work using big models to eliminate a dependency on manually-annotated concept labels. The proposed method proposes to construct a large-scale concept bank, which is used for predicting final labels. Some experiments demonstrate the effectiveness of the proposed method from different perspectives. The main concerns come from the technical novelty of this paper. All the reviewers were involved in the open discussion and its discussion point is still the novelty. Based on the results of the first round of comments, rebuttal, author-reviewer discussion, and reviewer-AC discussion, this paper could NOT be accepted for ICLR, due to limited novelty.

**Additional Comments On Reviewer Discussion:**

All the reviewers voted low rating scores of this paper in the first round, and, after rebuttal, only one reviewer raised his/her score to borderline acceptance while two reviewers lowered the score after discussion. In the reviewer-AC discussion phase, all the reviewers agree with the limited novelty of this paper.

---

### Decision · Program_Chairs · 2025-01-22

Reject